# Gene by Environment Interactions reveal new regulatory aspects of signaling network plasticity

**Matthew D. Vandermeulen**[ORCID]**, Paul J. Cullen**[ORCID]*

Department of Biological Sciences, University at Buffalo, Buffalo, New York, United States of America

* pjcullen@buffalo.edu

## Abstract

Phenotypes can change during exposure to different environments through the regulation of signaling pathways that operate in integrated networks. How signaling networks produce different phenotypes in different settings is not fully understood. Here, Gene by Environment Interactions (GEIs) were used to explore the regulatory network that controls filamentous/invasive growth in the yeast *Saccharomyces cerevisiae*. GEI analysis revealed that the regulation of invasive growth is decentralized and varies extensively across environments. Different regulatory pathways were critical or dispensable depending on the environment, microenvironment, or time point tested, and the pathway that made the strongest contribution changed depending on the environment. Some regulators even showed conditional role reversals. Ranking pathways' roles across environments revealed an under-appreciated pathway (**OPI1**) as the single strongest regulator among the major pathways tested (**RAS**, **RIM101**, and **MAPK**). One mechanism that may explain the high degree of regulatory plasticity observed was conditional pathway interactions, such as conditional redundancy and conditional cross-pathway regulation. Another mechanism was that different pathways conditionally and differentially regulated gene expression, such as target genes that control separate cell adhesion mechanisms (*FLO11* and *SFG1*). An exception to decentralized regulation of invasive growth was that morphogenetic changes (cell elongation and budding pattern) were primarily regulated by one pathway (**MAPK**). GEI analysis also uncovered a round-cell invasion phenotype. Our work suggests that GEI analysis is a simple and powerful approach to define the regulatory basis of complex phenotypes and may be applicable to many systems.

## Author summary

Phenotypes can change during exposure to diverse environments through the regulation of signaling networks. How networks produce different phenotypes in different environments is not fully understood. We explored the regulatory network that controls a complex phenotype in yeast by analyzing Gene by Environment Interactions (GEIs). GEI analysis showed that regulatory pathways can play different roles in different

**Data Availability Statement:** All data are in the manuscript and/or supporting information files.

**Funding:** MDV and PJC were supported by a grant from the NIH (GM098629). The funders had no role in study design, data collection and analysis,

decision to publish, or preparation of the manuscript.

**Competing interests:** The authors have declared that no competing interests exist.

environments, even reverse roles in specific contexts. Which pathway impacted phenotype the most changed across environments revealing decentralized control within the network. This finding led to the identification of an underappreciated pathway as a major regulator. Environmental differences caused changes in pathway interactions seen by conditional redundancy and conditional cross-pathway regulation. The environment also triggered conditional regulation of gene expression for target genes that were differentially regulated by different pathways. These insights into network plasticity may apply to signaling networks in general. Our work suggests that studying a biological process under diverse environments captures a more complete picture of its regulatory pathways. GEI analysis reduces the risk of mis-characterizing pathways studied in a single context and is a powerful method to explore the regulatory basis of complex phenotypes.

## Introduction

Phenotype is an essential feature of the identity and fitness of organisms. Many organisms show different phenotypes based on the environment (i.e. phenotypic plasticity). The ability to alter phenotypes enables organisms to acclimate to different settings and respond to stress. Phenotype is generated from genotype, the environment, and interactions between the two [Gene (or Genotype) by Environment Interactions (GEIs)]. GEIs are found across all kingdoms of life, including humans and other animals [1–6], plants [7–9], fungi [10–20], and bacteria [21–24]. Understanding how phenotypes are influenced by the environment is applicable to multiple areas of biology. GEIs play important roles in the production of livestock and crops [25–31], in organism adaptability to climate change [32–38], and in many diseases [39–45], including cancer [46–49].

One way that phenotypes can be regulated is by signal transduction pathways. Signaling pathways receive and process stimuli from external (or internal) environments and transmit information to trigger a response, usually by influencing the expression of target genes. Signaling pathways do not operate independently, and instead function with other pathways and protein complexes in signaling networks [50–54]. Such networks allow the integration of diverse stimuli into a response. Signaling networks have been extensively studied by protein-interaction tests, which reveal dozens to hundreds of interacting proteins [55,56], mathematical modeling to understand network fidelity and integration [57, 58], and increasingly by systems-wide approaches [59–61], which show the comprehensive nature of these web-like networks. Despite numerous advances in this area, many aspects of signaling network regulation remain mysterious. From the perspective of phenotypic plasticity, an interesting and important question is how do signaling networks change output phenotypes in different settings?

Here, we analyzed the regulatory network that controls filamentous/hyphal growth, which is a cell differentiation response that occurs in many fungal species including pathogens [62,63]. We focused on the budding yeast *Saccharomyces cerevisiae*, a model organism in which the response has been well characterized. During filamentous growth in yeast, round cells (i.e. yeast-form growth) switch to adhesion-linked "chains" of elongated cells [64] that can penetrate into substrates (i.e. invasive growth [65], **Fig 1A**). Invasive growth is induced by nutrient limitation (e.g. carbon or nitrogen), high cell density by sensing quorum-sensing molecules (e.g. ethanol), and changes in pH [66–70]. Invasive growth is inhibited by high osmolarity [71,72]. Many proteins required for invasive growth have been identified [73–75]. A subset of the proteins comprise evolutionarily conserved signaling pathways such as the filamentous

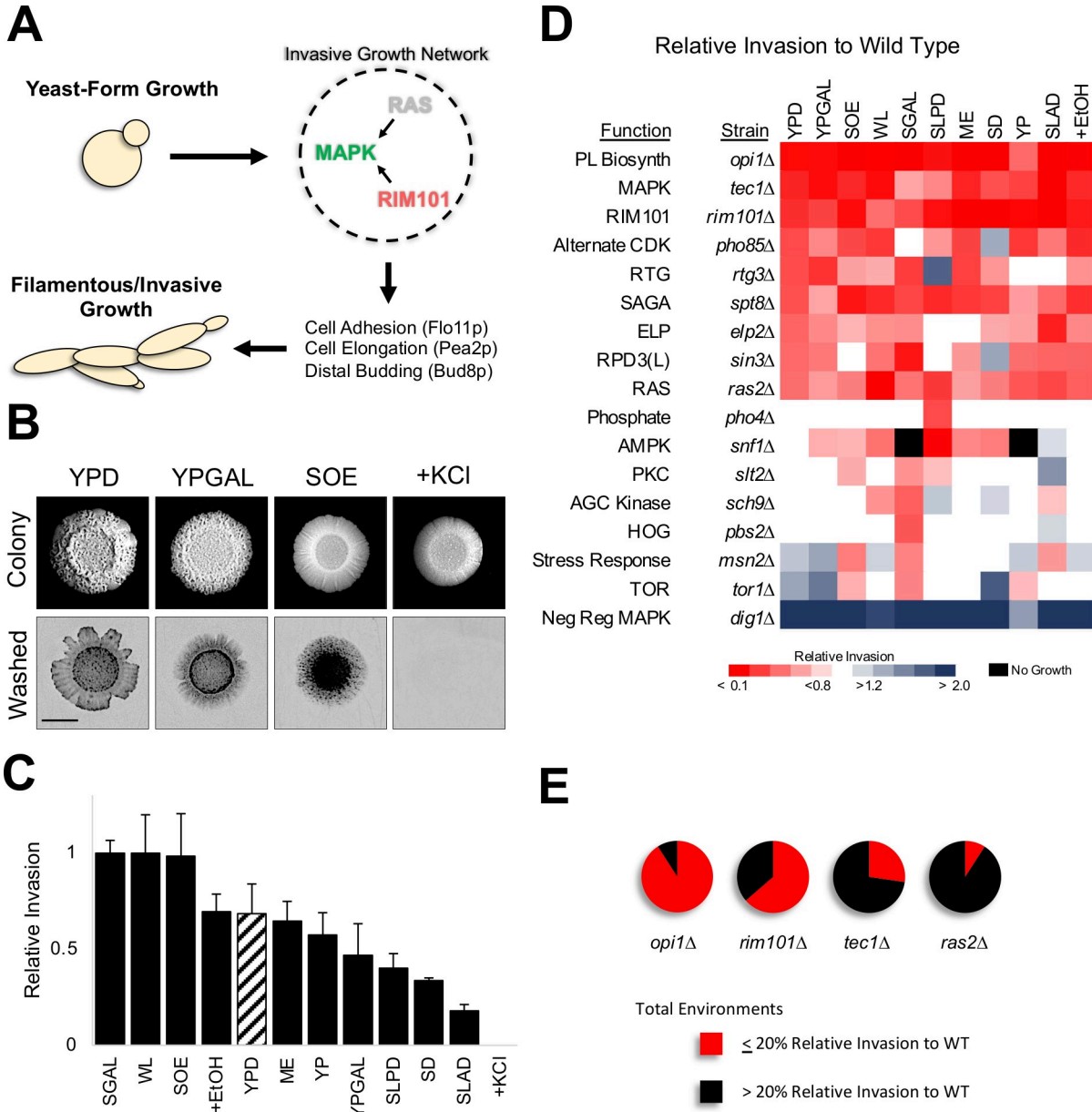

**Fig 1. The regulation of invasive growth varies across environments. A)** Yeast invasive/filamentous growth is regulated by pathways in a network (several major pathways are shown here), that can regulate each others' activity, and that regulate changes into the new cell type. **B)** PWA. Colony, cells before wash; Washed, invasive scars after wash, bar, 0.5 cm, inverted images are shown. **C)** Relative levels of invasive growth (values set to 1 for SGAL). Standard laboratory medium (YPD), diagonal lines. **D)** Heat map of relative invasion of network mutants compared to wild type by PWA. Vertical axis, strain. Horizontal axis, environments. White, same invasion as wild type, blue, more invasion, red, less invasion. Black, no growth. Raw data is shown in *S1 Fig*; Bar graphs with standard deviation in *S2 Fig*; and statistical tests that show significance of pairwise comparisons in *S1*–*S3 File*. The filamentous growth network includes: **OPI1**; **MAPK**; **RIM101**; **RAS**; **Pho85p** [alternate cyclin-dependent kinase (CDK)]; **RTG** (retrograde); **PKC** [Protein Kinase C pathway, regulates cell-wall integrity [226–228], and invasive growth [229]]; **AMPK** Snf1p, which derepresses the expression of enzymes that metabolize non-fermentable sugars [230–232] and positively regulates invasive growth [66]; **Pho4p**, a part of the phosphate regulon that responds to phosphate limitation [233–235]; **Msn2p**, a transcription factor involved in stress responses in yeast [236–238]; **HOG** (high-osmolarity glycerol), responds to high-osmolarity stress [71,72,239]; **TOR (**target of rapamycin), a highly conserved pathway in eukaryotes that regulates cell metabolism/growth/proliferation/ survival [240–243], invasive growth [244], and the **AGC kinase**, Sch9p [245]; **ELP** [the elongator complex [246–248], involved in regulating invasive growth [249]]; the chromatin-remodeling complex **Rpd3p(L)** [82,250], involved in regulating invasive growth [88]; and the chromatin-remodeling complex **SAGA** [Spt-Ada-Gcn5 acetyltransferase [251–253], involved in regulating invasive growth [86]]. **E)** Ranking of the main pathways was based on invasive growth compared to wild type (*S3 Fig*). The number of environments a mutant showed ≤ 20% (red) or > 20% (black) of wild-type invasion are shown.

growth Mitogen-Activated Protein Kinase (**MAPK**) pathway [65,66,76,77] and the cAMP-dependent Ras2p-Protein Kinase A (PKA) or **RAS** pathway [64,78–80], which both respond to limited glucose, and the **RIM101** pathway [67,81,82], which responds to changes in pH (**Fig 1A**). The signaling pathways that regulate invasive growth do not operate in isolation and are connected to each other at multiple levels to coordinate the filamentous growth response. Most regulatory pathways co-regulate a common set of target genes [83–87]. Key transcription factors regulate each other's expression [84], and several pathways regulate each others' activity [78,88–90] (**Fig 1A**, e.g. **RAS** and **RIM101** regulate **MAPK**), including at the levels of the protein kinases that comprise the pathways [91]. The filamentous/invasive growth network induces three major effector phenotypes during filamentous growth: an increase in cell-to-cell adhesion, a switch to distal-unipolar budding (from axial budding in haploids to promote growth away from other cells), and an increase in cell length [64,65,92] (**Fig 1A**). Numerous genes are involved in these effector phenotypes, including *FLO11*, which encodes the major cell adhesion protein [93–100], and whose expression is regulated by many transcription factors [79,84,85]; *BUD8*, which encodes a bud-site-selection protein that marks the distal pole [101–104]; and *PEA2*, which encodes a component of the polarisome, which is required for actin polymerization and cell elongation [105–110]. Thus, a highly integrated network of signaling pathways controls cell differentiation to filamentous growth in yeast.

Invasive growth has mainly been studied under select laboratory conditions. However, *S. cerevisiae* is a free-living microbe that encounters diverse environments in wild [111–114] and domesticated settings [115–119]. We reasoned that by exploring invasive growth in different environments, we might learn more about the regulatory basis of the response. By analyzing mutants on different environments (i.e. GEI analysis) in a broad-based survey, which is a useful approach to understand gene function [120,121], we gained new insights into the invasive growth regulatory network. Specifically, we found pathways made different contributions to the regulation of invasive growth in different environments. We were able to rank pathways by their overall contribution, which led to the identification of an underappreciated pathway as one of the main regulators of the response. We also show that depending on the environment, pathways showed conditional redundancy, role reversals, cross-pathway regulation, and conditional regulation of gene expression. We also identified a round-cell invasion phenotype. The knowledge gained from this simple approach may extend to signaling networks in general. Moreover, analyzing GEIs of signaling pathways may be a straightforward approach to employ in other systems to learn more about the regulation of complex phenotypes.

## Results

### GEI analysis uncovers diversity in invasive growth regulation and identifies a new major regulator

Invasive growth was examined across environments, including standard laboratory medium [YPD, yeast extract, peptone, dextrose], and media with varying amounts and/or sources of carbon, nitrogen, and phosphate, including media thought to mimic a natural environment [SOE, synthetic oak extract, [122]]. Media containing an inducer (ethanol) and inhibitor (KCl) of the response were also tested (13 environments total, *S1 Table*). Wild-type cells of the filamentous (Σ1278b) strain background were spotted onto the different media, and invasive growth was evaluated by the plate-washing assay (PWA, [65]), where a community of cells washed off a surface leaves a visible invasive scar (**Fig 1B**, YPD). Invasive growth was quantified and normalized to colony size for at least three biological replicates [M&M and [123]]. Wild-type cells showed a 10-fold range in invasive growth across environments (**Fig 1B and 1C**, full PWA in *S1A* and *S1B*, statistical analysis in *S1 File*). YPD medium induced a moderate

level of invasive growth (bar with cross hatch), while several environments induced robust invasion including the environment representing a natural setting (SOE). Invasive growth did not correlate with colony size (*S1C Fig*). These results support the existing idea that invasive growth is not binary (ON/OFF) but occurs in a phenotypic spectrum based on the environment. These properties make the invasive growth phenotype a strong candidate for GEI analysis.

The finding that invasive growth changes across environments indicates that the response may be regulated differently in different settings. Many signaling pathways, transcription factors, and protein complexes regulate invasive growth (**Fig 1D**, Function). To define how the regulation of invasive growth differs in each environment, the PWA was performed with mutants that perturb key pathways and protein complexes required for invasive growth (**Fig 1D**, Strain). Genes were chosen that ablate pathway activity based on previous tests comparing mutants that lack key pathway components [86,88,124]. The invasive growth of mutants was first compared to wild-type (**Fig 1D**, red, less invasive than WT; blue, more invasive, white, no difference, raw data *S1 Fig*, quantitation with standard deviation in *S2 Fig*, statistical analysis in *S2 File*). Most mutants showed differences in invasive growth from wild type (**Fig 1D**, e.g. YPD, tenth row, *ras2Δ* mutant showed decreased invasion relative to wild type, p-value <0.0001). Mutants also showed differences across environments (**Fig 1D**, compare colors across rows, statistical analysis in *S1 File*). For example, the *ras2Δ* mutant showed variable shades of red across environments (**Fig 1D**, e.g. compare YPGAL to WL, p-value < 0.0001, YPGAL to SLAD, p-value 0.04, and WL to SLAD, p-value 0.02). The mutants were also different from each other (**Fig 1D**, compare colors down columns, statistical analysis in *S3 File*). The *ras2Δ* mutant for example was not as critical as the *opi1Δ* mutant for invasive growth on YPGAL medium (**Fig 1D**, *ras2Δ* is a lighter shade of red, p-value 0.0006). These results demonstrate that the regulation of invasive growth changes based on the environment.

Because the role each pathway played in regulating invasive growth changed across environments, we asked whether some pathways play a major role in more environments than others, making them a stronger overall regulator of the biological process. To define the signaling pathways with the strongest overall role in regulating invasive growth, we ranked the regulators based on their roles in all environments collectively. We first averaged the invasive growth of each mutant relative to wild type across environments and arranged them in ascending order (*S3A Fig*). Because average relative invasion can have high variability, we also summed the number of environments that each mutant showed a specified reduction in invasion compared to wild type for several thresholds (*S3B Fig*). Both methods showed a different overall ranking of the regulators compared to the order seen in YPD (compare the order in **Figs 1D** to *S3*). Unexpectedly, both methods identified the transcriptional regulator Opi1p as the strongest single regulator of invasive growth (**Fig 1E**, *opi1Δ*). Opi1p is a positive regulator of invasive growth and biofilm/mat formation [125], and functions as a transcriptional repressor of phospholipid biosynthesis [126–128], but has not been appreciated as a major regulator of the response. Ranking also showed three well-established regulators of invasive growth among the top pathways: **RIM101** (by *rim101Δ*), **MAPK** (by *tec1Δ*), and **RAS** (by *ras2Δ*) [**Fig 1E**]. Because invasion phenotypes can be influenced by genetic background [129–131], our ranking may be specific to the strain background (Σ1278b) and environments tested here. The ability to order pathways allowed us to focus on the most relevant regulators. Thus, GEI analysis allowed ordering pathways that regulate invasive growth and identified an under-appreciated transcription factor as the major regulator of the response.

## Plasticity in invasive growth regulation revealed by GEI analysis between environments, microenvironments, and over time

To further explore variations in invasive growth regulation, strains lacking the main regulators (**RAS, MAPK, RIM101,** and **OPI1**), which showed no growth defects compared to wild-type cells (*S4 Fig*), were re-examined across environments. As shown above, **OPI1** was a major regulator in most environments (**Fig 2A–2C**, YPD, YPGAL, SLPD, WL, larger circle represents a stronger contribution). However, in one environment (YP), **OPI1** had a minor role (*S1 File*, p-value < 0.0001). The idea that a pathway can play a major role in some environments but not others was also true for **RIM101** (**Fig 2A–2C**, compare YP to WL, p-value 0.0001), **RAS** (**Fig 2A–2C**, compare WL to YPD, p-value 0.0005), and **MAPK** (**Fig 2A–2C**, compare YPGAL to SLPD, p-value < 0.0001). These results reveal a highly plastic regulatory network, where no single regulator has centralized control, and where different pathways play the major role in different environments.

We noticed when comparing two environments where wild type showed a change in invasion, some mutants showed the same change while other mutants showed no change. For example, wild-type cells showed increased invasion on WL medium compared to SLPD medium (**Figs 2A and 1C**, statistics in *S1 File,* p-value 0.0002) as did the *rim101Δ* mutant (**Fig 2A and 2B**, p-value 0.0005); however, the *opi1Δ* mutant showed a similar level of invasion (**Fig 2A and 2B**, p-value, n.s., 0.78). These data may suggest when a pathway is involved in sensing the difference between the environments (i.e. **OPI1** in this example) compared to a non-sensing role (i.e. **RIM101** in this example). Future studies aimed at testing these predictions by transitioning cells from one environment to another may uncover sensory mechanisms in the regulation of invasive growth.

Cells growing within microbial communities experience different microenvironments. Microenvironments can trigger specific responses in subsets of cells throughout the community [132]. Invasive growth occurs in distinct patterns, reflecting different microenvironments, that can be quantified and represented in a plot profile [123]. Plot profiles generate a representative cross section where less invasion (light pixel intensity) is represented by high values, and more invasion (dark pixel intensity) is represented by low values (**Fig 2D and 2E**, e.g. SGAL, wild-type, black line). Wild-type cells showed different invasion patterns in different environments (**Fig 2D and 2E**, black line, top and bottom panels). We examined how the major pathways might affect the regulation of invasive growth in different microenvironments. Cells lacking the main regulatory pathways also showed different patterns. For example, in one environment (SGAL), the *tec1Δ* mutant showed the highest invasion in the center of the scar, and the *ras2Δ* mutant showed the highest invasion in a surrounding ring (**Fig 2D and 2E**, SGAL, compare green and grey lines). This indicates that in SGAL medium, **RAS** is critical for invasion in the center region, and **MAPK** is critical in the surrounding ring region. In a different environment (SOE), the role of the regulatory pathways changed. On SOE, the *ras2Δ* mutant showed the most invasion in the center region, and the *tec1Δ* mutant showed a diffuse pattern (**Fig 2D and 2E**, SOE). This means that in some environments, the **MAPK** pathway is more critical than **RAS** for invasion in the center region (see also YPGAL, *S5 Fig*). These results support the idea that different pathways take on the major role in regulating invasive growth based on the environment.

Many pathways and effector genes in the invasive growth network also play a role in regulating complex-colony morphology [ruffling of cells above the surface [133]], which is an aspect of biofilm/mat formation [134]. Biofilm/mat formation also occurs in distinct patterns resulting from different microenvironments [135]. Wild-type colonies had a ruffled pattern on some environments (**Fig 1A,** YPGAL) but not others (**Fig 1A,** SOE). The different patterns did

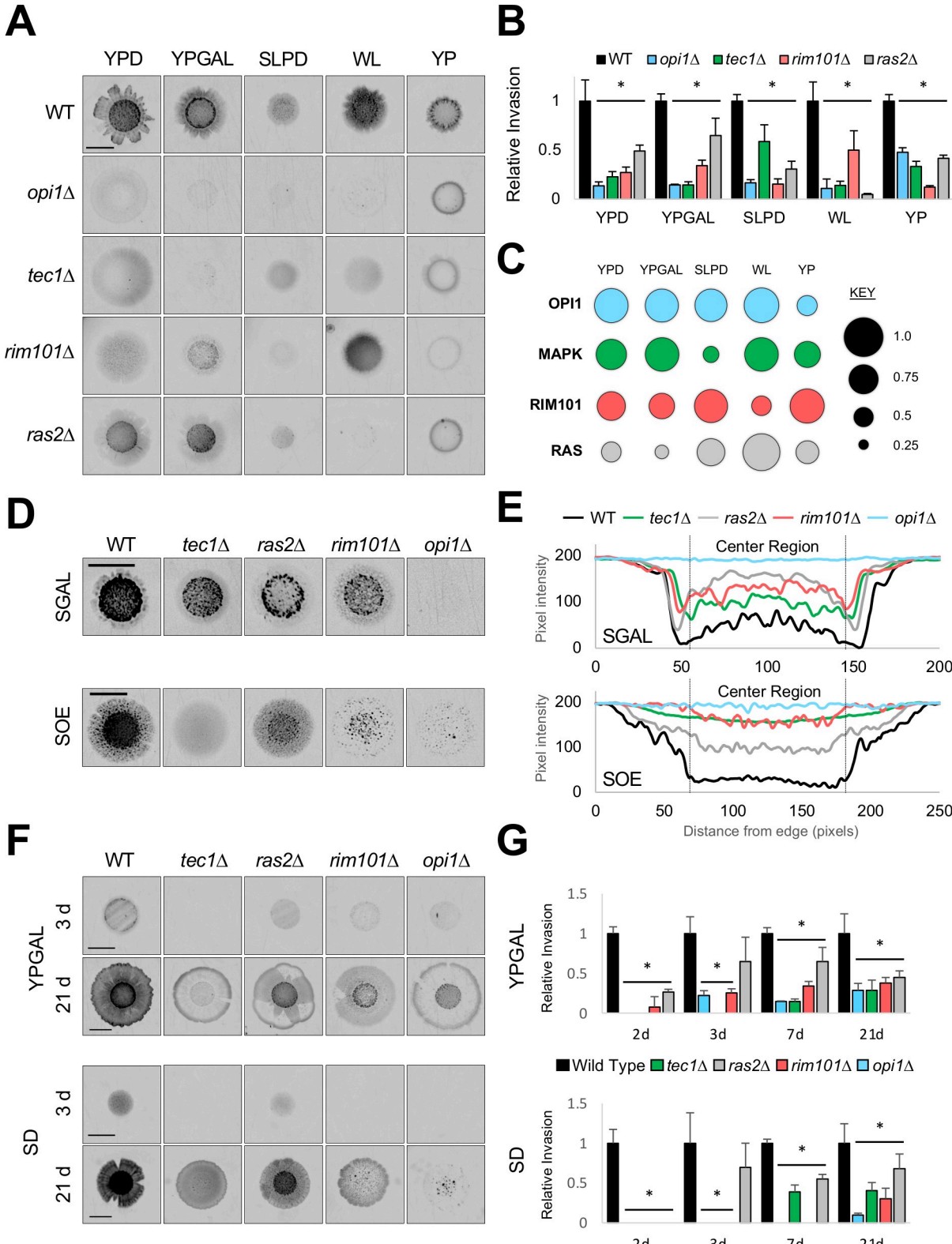

**Fig 2. Contribution of the main invasive growth regulators across environments, microenvironments, and over time. A)** PWA. Inverted images of invasive scars after wash, bar, 0.5 cm. Colonies prior to wash in *S1 Fig*. **B)** Levels of relative invasion to wild type, with wild type values set to 1. Asterisk, P-value ≤ 0.05, compared to wild type. **C)** Model of pathway contributions to invasive growth. Circle size is proportional to pathway contribution. **D)** PWA, see panel A for details. **E)** Plot profile of invasive growth across invasive scars. Strains and

media are as indicated; examples are shown in panel D. X-axis, distance (in pixels); Y-axis, pixel intensity. **F)** Invasive growth over time. PWA was performed at the times indicated. See *S7 Fig* for raw data. **G)** Invasive growth levels from panel F; see panel B for details. Asterisk, p-value ≤ 0.05, compared to wild type, by Student's t-test. YPGAL 7 d values are repeated from panel B.

not correspond to differences in invasive growth (**Fig 1B**, compare YPGAL to SOE, *S1 File*, p-value 0.0015). This result indicates that different regulatory mechanisms occur above and below the surface. Many pathways played a similar role in generating phenotypes above and below the surface (*S1 Fig*); however, at least one pathway played different roles in these micro-environments (*S6 Fig*, **RTG**). This result supports the idea that biofilm/mat formation and invasive growth are regulated by both overlapping and non-overlapping sets of pathways/genes [73]. Therefore, microenvironments reveal plasticity in the roles pathways play in regulating filamentous growth in different regions within a community of cells.

We became interested in understanding whether the pathways also differentially regulate invasive growth at different time points. To address this possibility, invasive growth was examined over time. In one environment (YPGAL), invasive growth increased over a three-week incubation (**Fig 2F**, full PWA in *S7A*). All four pathways played a major role in regulating invasive growth at an early time point tested (2d) but played different roles at different times (**Fig 2F and 2G**, YPGAL, 3d-21d). For example, **MAPK** was more critical at an early time point (3d) compared to a later time point (21d) for invasive growth (**Fig 2F and 2G**). Interestingly, the pathways made different contributions from one another at earlier time points (2d-7d) but made the same overall contribution at the last time point tested (21d) [**Fig 2G**, YPGAL]. However, the patterns were different (**Fig 2F**, YPGAL). Therefore, major regulatory pathways can make varying contributions during the progression of invasive growth.

GEI analysis was also used to explore the temporal development of invasive growth across environments. Like for YPGAL medium, on SD medium invasive growth increased over time (**Figs 2F** and *S7B*). However, pathways played different roles over time in the two conditions, based on the levels and patterns of invasive growth (**Fig 2F and 2G**, SD, and *S7C*). For example, when comparing 7d to 21d, **RIM101** contributed similarly to invasive growth at both time points in one environment (YPGAL) but differently at both time points in another environment (SD) [**Fig 2G**]. Moreover, all four pathways did not equally contribute by 21d on SD like they did on YPGAL medium (**Fig 2G**). Invasive growth after 21d also revealed GEIs for invasive growth patterns not seen at earlier time points between YPGAL and SD media (**Fig 2F**, 21 d, e.g. **RIM101** was required for center-ring invasion on SD but not YPGAL). Thus, the temporal roles pathways play in the development of the response may change in different environments.

## Additive and redundant roles of invasive growth regulators

When regulating a phenotype, signaling pathways can make separate (i.e. additive) contributions from one another, or they can make overlapping (i.e. redundant) contributions. To test if the major pathways played additive or redundant roles during invasive growth, double mutant combinations were generated among the major regulators. In most environments, the double mutants were less invasive than the single mutants. For example, on YPD, the *ras2Δ tec1Δ* double mutant was less invasive than the *ras2Δ* and *tec1Δ* single mutants [**Fig 3A and 3B**, full data set in *S8*)]. This indicates that the pathways play additive roles.

In some environments, double mutants showed the same invasion as single mutants. For example, the *ras2Δ tec1Δ* double mutant had the same level of invasive growth as the *ras2Δ* and *tec1Δ* single mutants on YP (**Fig 3A and 3B**). Also, the *ras2Δ opi1Δ* double mutant was less invasive than either single mutant in one environment (YPD), equal to one of the single

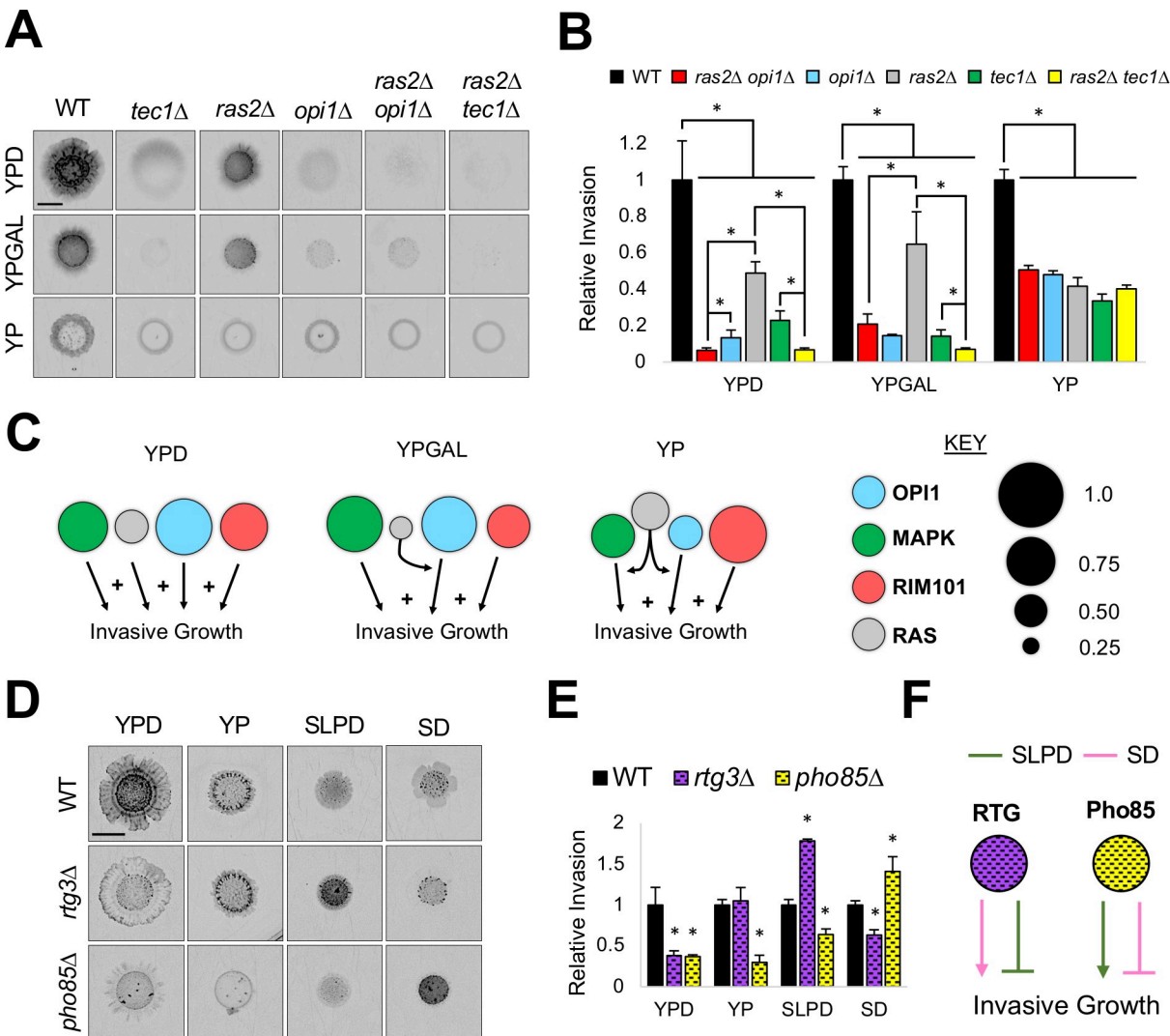

**Fig 3. Regulators of invasive growth show conditional redundancy and conditional role reversals. A)** PWA. Inverted images of scars after wash, bar, 0.5 cm. Colonies prior to wash in *S8 Fig*. **B)** Levels of relative invasion to wild type, with wild-type values set to 1. Asterisk, p-value ≤ 0.05, between the indicated strains by Student's t-test. **C)** Model of pathway contributions under different environments. Circle size is proportional to pathway contribution. (+) indicates an additive role. **D)** PWA, see Panel A for details. Colonies prior to wash in *S1 Fig*. **E)** Levels of relative invasion; details in panel B. Asterisk, p-value ≤ 0.05, compared to wild type by Student's t-test. **F)** Model of pathway contributions in different environments.

mutants in multiple environments (*opi1Δ* on YPGAL, SOE, WL), and equal to both single mutants in yet another environment (YP) [**Figs 3A, 3B** and *S8*]. This may suggest that a pathway in the network can make an additive contribution in one environment (**Fig 3C**, **RAS** on YPD), a redundant or similar contribution to one pathway in another environment (**Fig 3C**, **RAS** on YPGAL), and a redundant or similar contribution to more than one pathway in a third environment (**Fig 3C**, **RAS** on YP). In environments where double mutants invade, it can be inferred that at least one additional pathway is required. These observations are consistent with the idea that pathways that regulate invasive growth can play additive or redundant roles depending on the environment, which we refer to as conditional redundancy.

## Conditional role reversals between pathways can achieve similar invasive states

GEI analysis showed that several pathways promoted invasive growth in one environment and inhibited it in another (**Fig 1C**, **RTG**, **PHO85**, and other pathways, red or blue in different environments). These included Retrograde (**RTG**), which regulates the response to mitochondrial distress [136,137] and invasive growth [89,138,139]; and the alternate cyclin-dependent kinase (CDK) **Pho85p,** which controls cell-cycle progression in certain environments [140–143] and invasive growth [88]. These 'conditional role reversals' were explored in more detail. **RTG** was required for invasive growth in one environment (**Fig 3D** and **3E**, YPD, *rtg3Δ*) and inhibited invasive growth in another (**Fig 3D** and **3E**, SLPD). In some environments, **RTG** played no role (**Fig 3D** and **3E**, YP). **Pho85p** was required for invasive growth in SLPD and inhibited it in SD (**Fig 3D** and **3E**, *pho85Δ*). Intriguingly, when comparing SLPD to SD media, the environment where **RTG** played a negative role, **Pho85p** played a positive role and vice versa (**Fig 3F**). Despite these and other regulatory differences (*S9 Fig*), the amount and pattern of invasive growth for wild type was similar in both environments. Thus, wild-type cells can exhibit a similar phenotype between environments despite opposing roles for network regulators. **RTG** and **Pho85** might be functionally connected such that in the absence of one pathway, the other takes on a different role. In any case, this finding supports the idea that the same phenotype can arise through the action of pathways that have opposing roles in different settings.

## Signaling pathways conditionally regulate MAPK pathway activity across environments

Signaling pathways function in basal and activated states. In activated states, pathways can show different kinetics (amplitude, duration), which can result in different responses [ERK is a classic example [144]]. To define how the activity of the MAPK pathway might change across environments, wild-type cells containing a translational reporter whose activity reflects pathway activity [p*FRE-lacZ*, [89,145]] was examined. Based on reporter activity, the MAPK pathway showed a >10-fold range in activity across environments (**Fig 4A**). MAPK pathway activity correlated with invasive growth (*S7D Fig*). In line with previous findings [146], this result demonstrates that the MAPK pathway does not operate in an "ON/OFF" state but in a graded manor depending on the environment.

The levels of the reporter in cells lacking a positive regulator of the MAPK pathway defined a lower boundary for MAPK pathway activity (**Fig 4B**, *tec1Δ*, green). Reporter levels in cells lacking a negative regulator of the MAPK pathway, Dig1p [147–150], defined an upper boundary (**Fig 4B**, *dig1Δ*, purple). MAPK pathway activity was closer to the lower boundary in the environments tested (**Fig 4C**, MAPK pathway activity). Like MAPK pathway activity, MAPK-dependent invasive growth in wild-type cells was closer to the lower boundary (represented by the hypoinvasive *tec1Δ* mutant) compared to the upper boundary (represented by the hyperinvasive *dig1Δ* mutant [*S1* and *S2B Figs*]) in most environments tested, based on examination of invasive growth across conditions (**Fig 4C**, Invasive Growth) and over time (**Figs 4D** and *S7E*). One exception was on YP media, where invasive growth, but not MAPK pathway activity, was closer to the upper boundary (**Fig 4C**). This could be due to redundancy with another pathway that regulates invasive growth in this environment (i.e. RAS). Although an environment that triggers maximal invasive growth and MAPK pathway activity may remain to be identified, these results suggest that the MAPK pathway typically operates at less than half its maximum levels of activity, and promotes invasive growth to less than half its maximal levels, even in

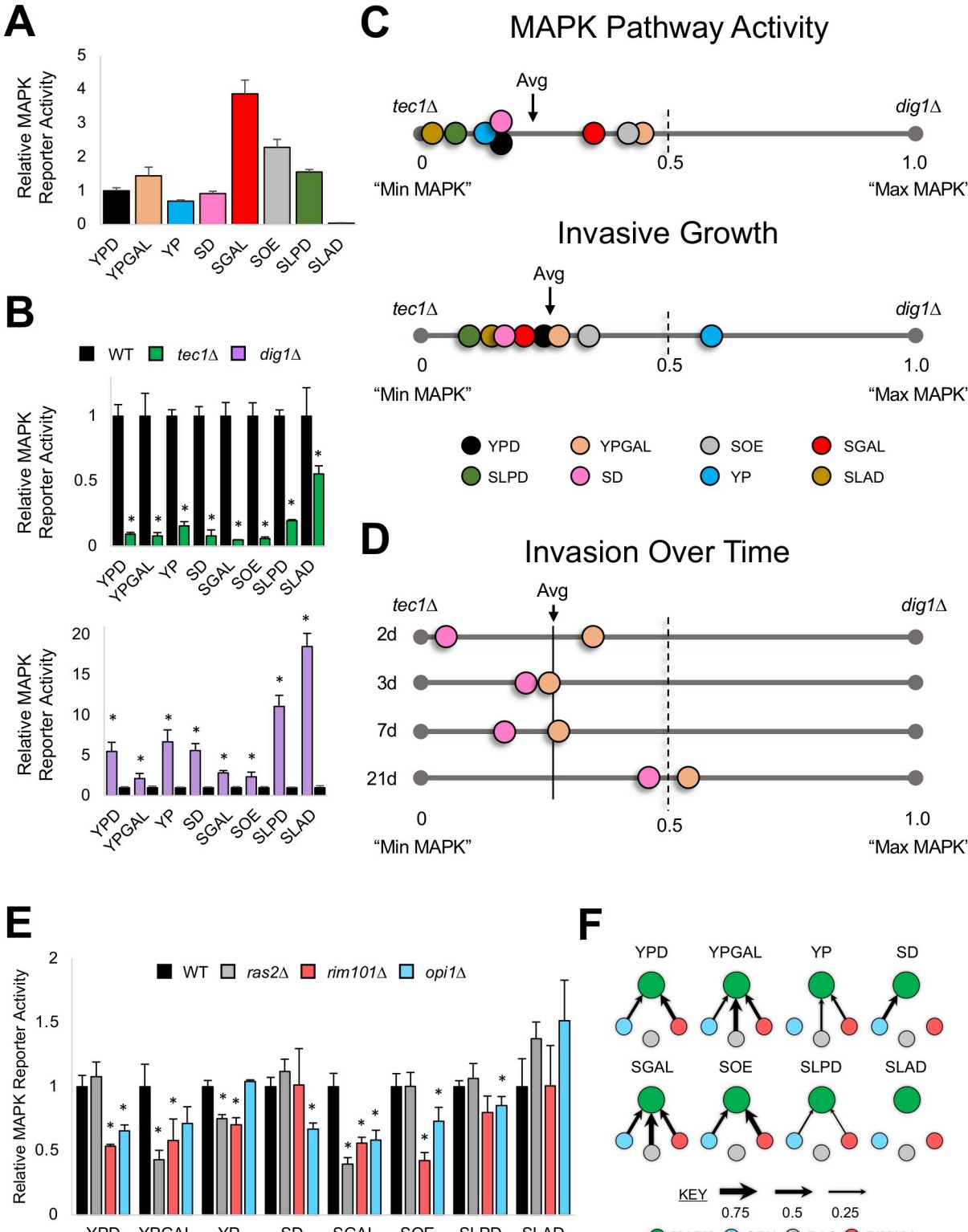

**Fig 4. MAPK pathway activity is curbed and is conditionally regulated by other major invasive growth regulators. A)** ß-galactosidase (*lacZ*) assay. Wild-type cells were quantified for **MAPK** pathway activity by the p*FRE-lacZ* reporter in indicated media. Values were normalized to YPD, which was set to 1. **B)** ß-galactosidase (*lacZ*) assay. Wild-type cells and the *tec1Δ* and *dig1Δ* mutants were quantified for relative **MAPK** pathway activity. Wild-type values were set to 1. Top panel, *tec1Δ*, green. Bottom panel, *dig1Δ*, purple. Asterisk, p-value ≤ 0.05, compared to wild type. **C)** Graph of **MAPK** pathway activity or Invasive Growth for wild-type cells compared to the *tec1Δ* (set to zero) and

*dig1Δ* (set to 1.0) mutants. The average **MAPK** pathway activity or Invasive growth of wild-type cells across environments is marked (Avg arrow). **D)** Same as panel C, except for invasive growth at indicated time points (2d, 3d, 7d, 21d). **E)** ß-galactosidase (*lacZ*) assay. Indicated strains were quantified for relative **MAPK** pathway activity, with wild type values set to 1. Asterisk, p-value ≤ 0.05, compared to wild type. p-value, 0.086 for the *opi1Δ* mutant in YPGAL and 0.07 for the *rim101Δ* mutant in SLPD. **F)** Pathway contributions to **MAPK** pathway activity in different environments. Arrow size is proportional to pathway contribution.

environments where the pathway is activated to stimulate invasive growth. Thus, the MAPK pathway functions at modest levels to induce invasive growth.

Signaling pathways can regulate each other's activity [151–156]. In the filamentous/invasive growth network, **RAS** [78,89], **RIM101** and **OPI1** [89], regulate the activity of the **MAPK** pathway. Therefore, we became interested in understanding whether these pathways regulate **MAPK** pathway activity differently across environments. To test if the role in cross-pathway regulation of **RAS**, **RIM101**, or **OPI1** changes, **MAPK** pathway activity was examined in pathway mutants across environments. Each pathway conditionally regulated **MAPK** pathway activity (**Fig 4E and 4F**, arrow size represents strength of contribution). For example, in one environment, all three pathways regulated the **MAPK** pathway (**Fig 4E and 4F**, SGAL). In other environments, the pathways did not: on YPD medium **RAS** was not required (**Fig 4E and 4F**); on YP medium, **OPI1** was not required (**Fig 4E and 4F**); and on SD medium, **RIM101** was not required (**Fig 4E and 4F**). The fact that **RAS, RIM101**, and **OPI1** contribute to **MAPK** pathway activity in some environments but not others demonstrates that environment controls cross-pathway regulation in the signaling network. This offers one mechanism for how **MAPK** pathway activity (and possibly invasive growth) exhibits plasticity. Furthermore, conditional cross-pathway regulation may account for how conditional redundancy occurs between the **RAS** and **MAPK** pathways (**Fig 3C**).

**MAPK** pathway activity might be conditionally regulated by altering the expression of the genes encoding pathway components. The sensor of the pathway, *MSB2*, the kinase of the pathway, *KSS1*, and the transcription factor, *TEC1*, show different expression patterns in different environments [83,86]. Moreover, global gene expression analysis and protein-DNA interaction tests showed **RIM101** regulates *MSB2* [89] and *TEC1* expression [157]; **OPI1** regulates *MSB2* [89] and *KSS1* expression [157]; and **RAS** regulates *MSB2* [84,85,89] and *TEC1* expression [84] (*S2 Table*). Other pathways might regulate each other's activities as well. The expression of the *RAS2* gene is regulated by **MAPK** [158], **RIM101** [159], and **OPI1** [157]. The expression of the gene encoding the **RAS** pathway transcription factor, *FLO8*, is regulated by **MAPK** [85,160] and **OPI1** [161]. The expression of the *RIM101* gene is regulated by **RAS** [85,157] and the expression of *RIM8* is regulated by **MAPK** [89,160]. The expression of *OPI1* is regulated by **RIM101** [159]. Thus, pathways within the network may regulate each other's activities by regulating genes encoding pathway components. The important conclusion from this section is that GEI analysis can reveal new insights into the activity and regulation of signaling pathways that function in an interconnected network.

## Pathways differentially regulate separate cell adhesion target genes

Upstream signaling events induce three well-established effector phenotypes during filamentous growth: an increase in cell-to-cell adhesion, a switch to distal budding, and an increase in cell length [64,65,92]. The three effector phenotypes have some independence from each other in regulating invasive growth [162,163]; however, whether adhesion, distal budding, and cell elongation change their contribution to invasion across environments is unknown.

To explore how effector processes contribute to invasive growth in different environments, mutants that compromise each aspect [*flo11Δ* (reduced adhesion), *bud8Δ* (reduced distal

budding), and *pea2Δ* (reduced elongation)] were tested by the PWA. We found each aspect of invasive growth contributed differently from one another, in agreement with previous studies [162,163], and their contribution changed based on the environment (**Fig 5A**, full data set in *S1-S2*, statistical analysis in *S1–S3 File*). *FLO11*-dependent adhesion played a major role in invasive growth in most environments (**Fig 5A–5C**), but not all (YPGAL). Similarly, distal budding and cell elongation played major roles in several environments (**Fig 5A–5C**, e.g. YPGAL and SOE) and minor roles in others (e.g. YP and SGAL). Each aspect behaved differently than the other aspects in at least one environment (**Fig 5D**, *PEA2* on YPD, *FLO11* on YPGAL, *BUD8* on +EtOH). The *flo11Δ*, *bud8Δ*, and *pea2Δ* mutants also showed different patterns of invasion. On SOE medium, for example, the *flo11Δ* mutant showed pinpoint invasion, while *bud8Δ* and *pea2Δ* mutants showed uniform invasion (**Fig 5B**). Therefore, one explanation for the diversity in invasive growth induced by the signaling network comes from differences in the roles of effector processes.

Because adhesion, distal budding, and elongation affect invasive growth differently, we hypothesized the differential regulation of these aspects by signaling pathways could account for how the regulatory network controls the variation in invasive growth seen in different environments. To examine the roles of pathways in regulating cell adhesion independent of the other aspects of filamentous growth, the size of groups or clumps of adhesive cells in pathway mutants was measured during growth in liquid cultures as described [123]. Signaling pathway mutants showed reduced adhesion compared to wild type and different levels from each other (**Fig 5E and 5F** and *S10*). For example, **RIM101** was the main regulator of adhesion in YPGAL and SOE medium compared to the other pathways (**Fig 5F**). Furthermore, the pathways showed different levels of adhesion across environments. For example, **OPI1** played a stronger role in SD than YPD media (**Fig 5F**). Thus, signaling pathways show differences in cell adhesion in different contexts, which suggests the pathways differentially regulate cell adhesion based on the environment. This conclusion was supported by examination of complex colony morphology (*S11A Fig*), which is also regulated by the same pathways and Flo11p [133,135,164]; within-colony adhesion [by a method we developed [123], *S11B Fig*]; and adhesion of cells to plastic (*S12 Fig*), another Flo11p-dependent adhesion phenotype [134,165].

**MAPK**, **RAS**, **RIM101**, **OPI1,** and other pathways regulate *FLO11* expression [79,82,84,85,92,125], which has one of the largest promoters in the genome and is a site at which multiple transcription factors converge. Not surprisingly, the *tec1Δ*, *rim101Δ*, *ras2Δ*, and *opi1Δ* mutants showed reduced *FLO11* expression by quantitative reverse transcription PCR (RT-qPCR) analysis (**Fig 5G**, left, YPD). In line with our hypothesis, the pathways regulated *FLO11* expression to different levels from each other. In YPD, **RIM101** and **OPI1** played the major roles (>10-fold), followed by **MAPK** (3.3-fold), and **RAS** (1.3-fold). One pathway (**RAS**) showed a GEI in regulating *FLO11* expression across environments (**Fig 5G and 5H**, compare 2-fold change in YPD to YPGAL).

Unexpectedly, some pathways played a larger role than Flo11p in several environments. The *tec1Δ* and *opi1Δ* mutants showed less invasion (e.g. YPGAL, *S1* and *S2 Figs*, *S3 File*) and the *tec1Δ*, *ras2Δ*, *rim101Δ*, and *opi1Δ* mutants were less adherent (**Fig 5E and 5F**) than the *flo11Δ* mutant in several environments. These data reveal a Flo11p-independent adhesion mechanism that is regulated by pathways in the network. We previously showed that the **MAPK** pathway regulates the expression of *SFG1* [83,86,160], which encodes a transcription factor involved in the regulation of filamentous growth [166]. Sfg1p, along with other transcription factors [167–169], regulates daughter-cell separation [170], has a minor role in regulating the expression of *FLO11* [123], and mainly contributes to invasive growth by an adhesion mechanism that is separate from Flo11p [123]. Along with **MAPK**, the **RAS, RIM101**, and **OPI1** pathways regulated *SFG1* gene expression (**Fig 5G,** right, YPD). Thus, the main pathways that regulate filamentous

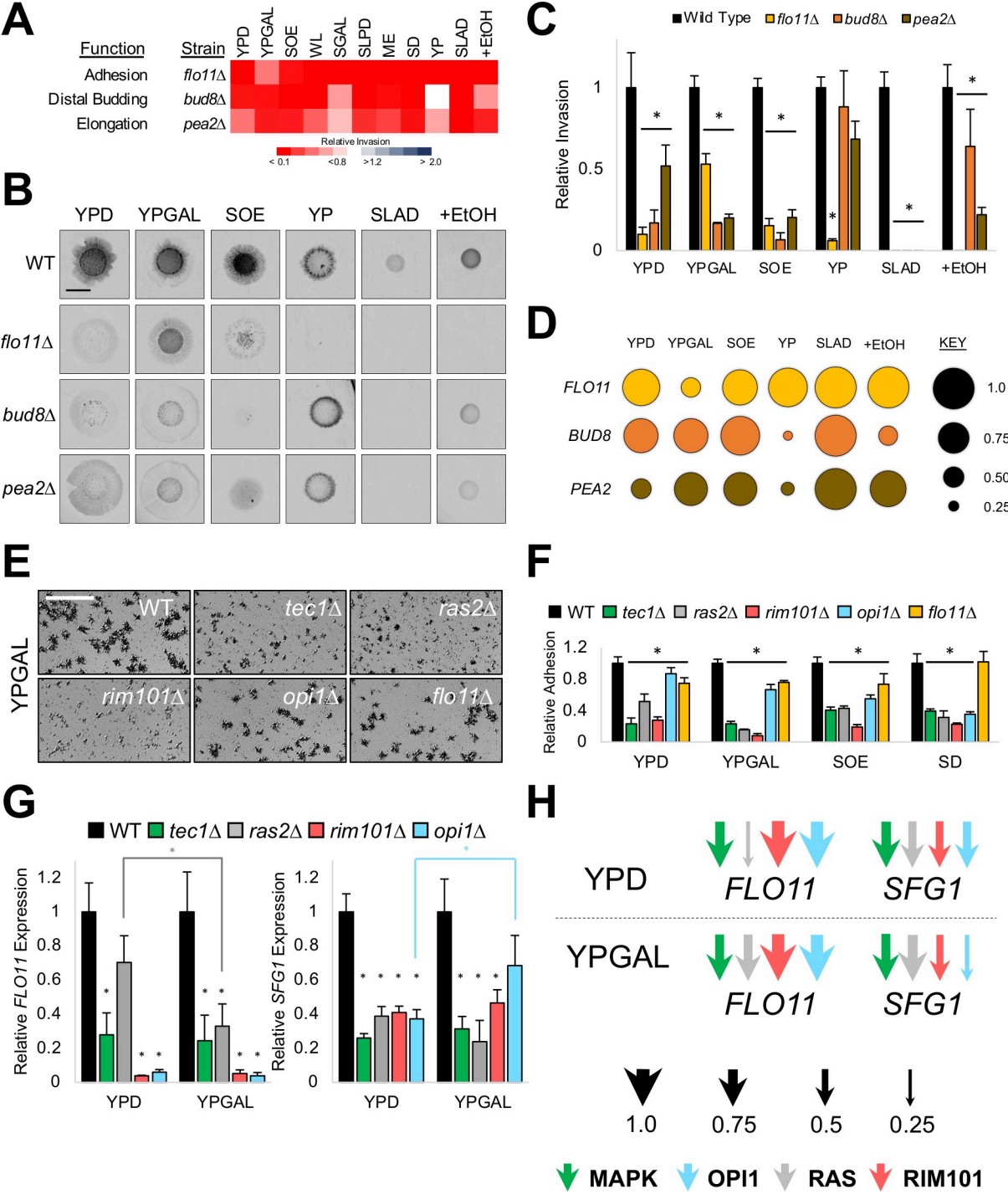

**Fig 5. Effector process regulation by the main pathways includes conditional expression of adhesion target genes A)** Heat map of relative invasion of indicated mutants compared to wild type by PWA. See **Fig 1B** for details. **B)** PWA; Inverted images of invasive scars after wash, bar, 0.5 cm. Colonies prior to wash in *S1 Fig*. **C)** Levels of relative invasion, with wild type values set to 1. Asterisk, p-value ≤ 0.05, compared to wild type. **D)** Model of effector contributions to invasive growth. Circle size is proportional to effector contribution. **E)** Cell adhesion in liquid cultures. Cells were imaged by microscopy at 5X magnification, bar, 200 μm. All environments in *S10 Fig*. **F)** Quantification of relative adhesion by average area of cell clusters, normalized to wild type, set to 1. Asterisk, p-value ≤ 0.05, compared to wild type. **G)** RT-qPCR analysis of mRNA levels (left, *FLO11*; right, *SFG1*) between indicated strains and environments. Wild-type values were normalized to *ACT1* expression and set to 1. Black asterisk, p-value < 0.05, compared to wild type. Grey asterisk, p-value < 0.05, for *FLO11* expression in the *ras2Δ* mutant between YPD and YPGAL. Blue asterisk, p-value < 0.05, for *SFG1* expression in the *opi1Δ* mutant between YPD and YPGAL. **H)** Model of pathway contributions to *SFG1* and *FLO11* gene expression. Arrow size is proportional to the pathway contribution.

growth regulate this cell adhesion mechanism. Support for this conclusion comes from large-scale analysis showing that **RIM101** regulates *SFG1* expression through the transcription factor Rim101p by gene expression profiling [171] and chromatin immunoprecipitation (CHIP) analysis [172]]. **OPI1** also regulates *SFG1* by the transcription factors Ino2p by microarray analysis [158] and Ino4p by CHIP [173]. **RAS** regulates *SFG1* expression [83,86], perhaps through the transcription factors Flo8p, Sok2p, and Phd1p, which bind by CHIP [84].

*SFG1* (like *FLO11*) showed changes in expression by signaling pathways across environments. In particular, when considering the mutant analysis, **OPI1** showed a > 1.7-fold change between YPD and YPGAL media in regulating *SFG1* gene expression (**Fig 5G and 5H**). Comparing target gene expression also showed that **RIM101** and **OPI1** were more critical for *FLO11* expression compared to *SFG1* (**Fig 5G and 5H**). By comparison, **RAS** was more critical for *SFG1* expression than *FLO11* (**Fig 5G and 5H**), and the **MAPK** pathway showed equal regulation of both (**Fig 5G and 5H**). These results collectively demonstrate that the main pathways that control invasive growth control the expression of target genes that control separate cell adhesion mechanisms. Other targets that regulate invasive growth are most likely changing conditionally, because some **MAPK** pathway targets are regulated by the **MAPK** pathway in one environment but not another (*S13 Fig*). Such regulation may account for the phenotypic plasticity of invasive growth.

## The MAPK pathway is the main regulator of cell elongation and distal budding during invasive growth

Like for adhesion, the roles pathways play in regulating cell elongation and distal budding may also differ between pathways. The **MAPK** pathway is known to regulate cell elongation by extending the G1 phase of the cell cycle [174] by regulating expression of *CLN1* [160,175], which results in prolonged growth at the tip of the cell. The role of the other pathways in regulating cell elongation have not been fully explored. Measuring the length of filamentous cells showed that **RIM101** and **OPI1** played no role in regulating cell elongation, **RAS** played a minor role, and **MAPK** played the major role (**Fig 6A**, L/W; see additional environments in *S14*). A similar result was obtained for distal-pole budding: **MAPK** was the major pathway to regulate distal budding, while the other pathways played minor roles (**Fig 6A**, CFW, DB%). These results indicate that the **MAPK** pathway, and to a lesser degree **RAS**, regulate differentiation to the filamentous cell type.

The fact that strains that cannot produce elongated cells exhibit invasive growth on most environments (*S1* and *S2* Figs, *tec1Δ* and *pea2Δ*) indicates that cell elongation is dispensable for invasive growth. Previous reports have suggested that cells with a low level of filamentation can exhibit invasive growth [176]. To further investigate this phenotype, the invasive scar of the *tec1Δ* mutant was inspected by microscopy. Like for wild-type cells, the *tec1Δ* mutant invaded below the agar surface (**Fig 6B**, close up, white arrows) despite the fact that cells were round in appearance (**Fig 6B**, close up, compare yellow arrows). Cross sections of the invasive scar showed sub-surface invasion (**Fig 6B**, cross sections, full imaging in *S15*) and agar squashes of embedded invasive cells revealed a round-cell morphology (*S16 Fig*). GEI analysis can therefore uncover new phenotypes, like the round-cell invasion phenotype described here.

## Discussion

Here we employed GEI analysis in a model genetic organism to explore the regulation of a complex phenotype. We hypothesized new insights would be gained by exploring several environments because invasive growth is typically studied under select laboratory conditions. Our approach uncovered new insights into the network that regulates invasive growth. It allowed

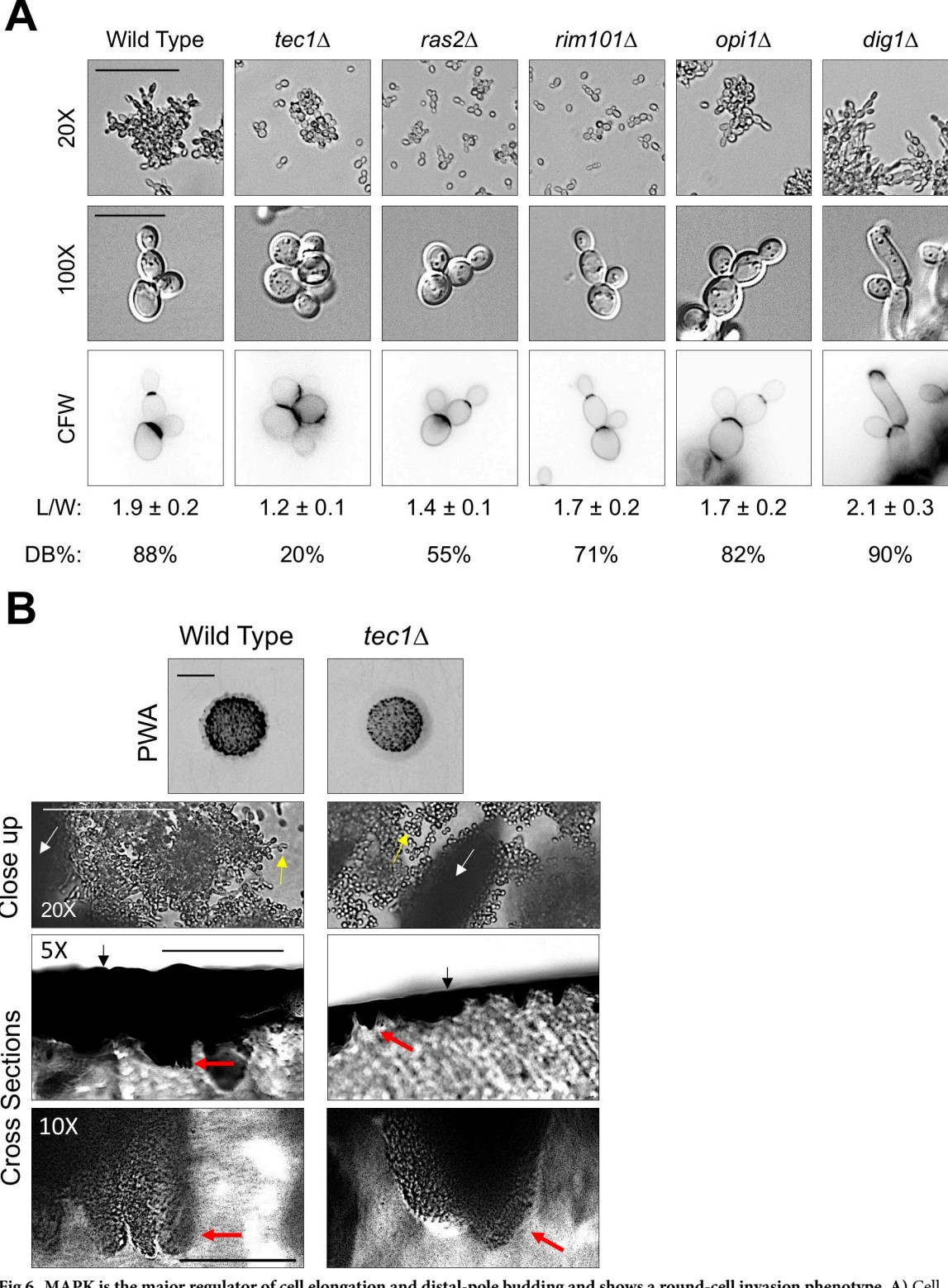

**Fig 6. MAPK is the major regulator of cell elongation and distal-pole budding and shows a round-cell invasion phenotype. A)** Cell morphology analysis. Cells were compared by microscopy at 20X and 100X magnification after growth in YPGAL for 21 h, 20X bar, 50 μm, 100X Bar, 10 μm. CFW, calcofluor white stained images of bud scars. L/W, quantification of cell length-to-width ratio. DB%, percentage of cells with distal-budding pattern. **B)** Cross section analysis. PWA, inverted image of invasive scar, bar, 0.5 cm. Close up, 20X magnification image by microscopy of invasive scar, bar, 100 μm. White arrows, invading cells below the agar surface (out of focus).

Yellow arrows, show elongated morphology for wild type compared to round-cell morphology in the *tec1Δ* mutant. Cross sections, top row, 5X magnification by microscopy of invasive scar cross section, bar, 500 μm. Black arrows, agar surface. Red arrows, invading cells. Bottom row, 10X magnification by microscopy of invasive scar cross section, bar, 100 μm. Red arrows, same invading cells as 5X magnification.

us to rank the pathways in the network and assign an underappreciated regulator as a new major regulator; it uncovered conditional role reversals for some regulators; it revealed new aspects of regulation occurring between pathways in the network, such as conditional pathway redundancy and cross-pathway regulation; it revealed the conditional regulation of gene expression for downstream targets of the signaling network; and it identified a new phenotype (round-cell invasion). Given that many of the signaling pathways that regulate invasive growth are evolutionarily conserved, these finding may be broadly applicable to signaling networks in general.

A specific benefit to GEI analysis is the ability to rank the contributions of regulatory pathways to a biological response. For the filamentous/invasive growth network, GEI analysis allowed identification of the major regulators distinguishing them from pathways that contribute only under select environments. Ranking led to the unexpected discovery of an underappreciated transcriptional repressor (**OPI1**) as one of the major regulators of the response. Opi1p is a transcriptional repressor of phospholipid biosynthesis and inositol production [126,127,177,178]. As such, **OPI1** controls the switch between lipid storage, which occurs during fasting, and membrane production, which occurs during growth [179,180]. Opi1p is inactivated by binding directly to phosphatidic acid [128,181,182], which is a conserved signaling molecule in plants and mammals [183–186] and whose misregulation is associated with cancer and other diseases [187–192]. Thus, analogous phospholipid regulatory circuits may regulate eukaryotic regulatory networks to control morphogenetic pathways (e.g. **MAPK**) in healthy and disease states.

GEI analysis also allowed interrogation of the flexible and invariant components of the network and showed that the major pathways (**MAPK**, **RAS**, **RIM101**, and **OPI1**) had different roles in different environments, microenvironments, and over time. Most pathways were at least partially dispensable under one or more environments revealing unexpected network plasticity. Several pathways showed conditional redundancy with other pathways. Conditional redundancy implies pathway interactions within the network are conditional and may apply to all types of proteins in networks. Perhaps this may suggest redundancy in protein function is not conserved for robustness or genetic buffering, but instead is conserved because non-redundant functions exist in some environments. In line with the finding of conditional redundancy, cross-pathway regulation was conditional: **RAS**, **RIM101**, and **OPI1** regulated **MAPK** pathway activity only under select environments. Collectively, these results indicate that none of these pathways function as the central regulator and that the lead role is environmentally determined. Decentralizing control of invasive growth could be advantageous because it allows multiple pathways that sense different stimuli to take the leading role in different settings.

Surprisingly, conditional role reversals occurred for some pathways that control invasive growth. In these instances, GEI analysis reduces possible mischaracterization of a pathway's role due to the exploration on multiple environments. For example, if **RTG** was only explored on one condition (e.g. SLPD), it might be labeled as a negative regulator, even though in most environments, that pathway plays a positive role. There is precedent for transcription factors switching from positive to negative roles, which can occur in different nutrient states [193]. However, here we show that two pathways (**RTG** and **PHO85**) play opposing roles under

different conditions to create a similar phenotype. How this type of regulation occurs is not clear. It will be interesting to see if conditional role reversals occur for one or all phenotypes a pathway regulates.

The fact that **MAPK** pathway activity functions at modest levels to induce invasive growth is consistent with previous results from our laboratory. The **MAPK** pathway induces expression of genes encoding multiple negative regulators of invasive growth and a negative regulator of **MAPK** pathway activity (*NFG1*), which has the effect of modulating the response [123]. Curbing invasive growth is advantageous in the balance between invasion and colony expansion [100]. Curbing **MAPK** pathway activity may be advantageous in that pathway hyperactivation leads to defects in budding and cell morphology [88,146,194]. Perhaps other pathways operate at low overall levels as a safeguard to curb pathway activity. It is clear at least for the mammalian RAS-MEK-ERK pathway that hyperactivation can lead to cancer and other diseases [195].

Many pathways independently converge on the *FLO11* promoter. We show here that *SFG1* is regulated by the major filamentation regulatory pathways as well. At least two pathways conditionally regulate the *SFG1* and *FLO11* promoters, which suggests that each of the major pathways independently regulates each adhesion target. Combinatorial control by multiple pathways regulating multiple targets that control adhesion by independent mechanisms can explain much of the diversity in invasive growth seen across environments. It will be interesting to determine how transcription factors bind to the *FLO11* and *SFG1* promoters, and we predict, based on GEI analysis, that they may bind to these promoters independently of each other and differently at the different promoters. This could occur by the transcription factors having different binding affinities or by differences in chromatin structure/organization between the promoters. The importance of cell adhesion in the regulation of filamentous/invasive growth can be seen from the existence of multiple adhesion mechanisms and the growing complexity of the adhesion code. The regulatory network may also regulate the expression of other adhesion molecules [*FLO1*, *FLO10*, *FLO9*, *FLO5* [163,196–198]] and cell-wall degrading enzymes that control cell separation [*SCW11*, *DSE1/2/4*, *CTS1*, *ENG1* [167–169,199]] in an environment-dependent manner.

An exception to decentralized control of the regulatory network is that the **MAPK** pathway is the central regulator of polarity and cell shape. Because differentiation requires alterations to the cell cycle to trigger cell elongation [174], it may be beneficial to employ a single pathway as the main regulator. This finding led to the identification of a round-cell invasion phenotype. Round-cell invasion could have important ramifications to understanding virulence if conserved in pathogenic species. Although filamentous growth is essential for virulence in *Candida albicans* and *Candida glabrata* [200–205], how cells penetrate the host is not completely understood. Moreover, given that *C. albicans* contains even more cell adhesion molecules than *S. cerevisiae*, it may be instructive to use GEI analysis to uncover new facets of this dimorphic fungal response.

Filamentous growth was regulated at different levels in microenvironments across the invasive scar and above and below the agar surface. This may reflect differences in pH or glucose concentrations in the different environments, which exist in gradients from the center of a community of cells to the outside rim [135]. This conclusion could have important ramifications, because microenvironments play a role during infections by *C. albicans*, where intruders move from the site of infection to new tissues [206–210]. Microenvironments also play a major role in diseases like cancer [57,211–213], including cancer cell invasion into new environments [214]. Intriguingly, it has recently been speculated that tumor cells encountering a new microenvironment might alter cancer progression independent of additional mutations

[215]. Therefore, GEI analysis may be a useful tool to interrogate signaling networks that control complex phenotypes involved in disease and infection.

In summary, GEI analysis of a complex phenotype has revealed unexpected features of plasticity in an evolutionarily conserved regulatory network. Insights uncovered here may apply to orthologous pathways in fungal pathogens and to eukaryotic signaling pathways in general. GEI analysis can reveal new insights (and new phenotypes) by capturing the different environmental roles of regulatory pathways and preventing the characterization of only environment-specific functions. The knowledge gained from applying this simple approach demonstrates the advantages of looking at biological systems across diverse environments and suggests this approach will be useful in interrogating other systems where signaling networks govern complex phenotypes.

## Materials and methods

### Yeast strains and plasmids

Yeast strains are listed in *S3 Table*. All strains are isogenic with HYL333 of the Σ1278b background [provided by G. Fink [76]]. Homologous recombination was used to make gene deletions using auxotrophic/antibiotic resistance markers amplified by polymerase chain reaction (PCR). Templates were introduced into yeast by lithium acetate transformation as described [216]. PCR southern analysis and phenotype, when possible, were used to verify strains. p*FRE-lacZ* plasmid is used to measure the transcriptional level of the MAPK pathway as described in [217].

### Media

Variations of YP based media were prepared as follows: YP: 1% yeast extract, 2% peptone; YPD: YP with 2% dextrose; YPGAL: YP with 2% galactose; +KCl: YPD with 1M KCl; and YPD (High Glu): YP with 10% dextrose. Synthetic media was prepared as follows: SD: 0.67% yeast nitrogen base without amino acids, amino acids, 2% dextrose; SGAL: 0.67% yeast nitrogen base without amino acids, amino acids, 2% galactose; SLAD [synthetic low ammonium, [64]]: 0.17% yeast nitrogen base without amino acids or ammonium, 2% dextrose; +EtOH: SLAD with 2% ethanol [70,218]; and SLPD [synthetic low phosphate, adjusted from [219]]: 0.17% yeast nitrogen base without amino acids/phosphates/NaCl, 0.1% KCl, 0.01% NaCl, amino acids, 0.01% $KH_2PO_4$, 2% dextrose. Natural and domesticated environment mimics were prepared as follows: SOE [synthetic oak extract [122]]: 1% sucrose, 0.5% fructose, 0.5% glucose, 0.1% yeast extract, 0.15% peptone; and ME [Malt extract medium, recipe from Teknova (https://www.teknova.com/malt-extract-broth)]: 0.6% malt extract, 0.18% maltose, 0.6% dextrose, 0.12% yeast extract. WL: Wallerstein laboratory nutrient agar [220] [ordered from MilliporeSigma cat# 17222]. Media was supplemented with uracil for auxotrophic markers when applicable. For solid media, 2% agar was added, except SOE which had 1.6% agarose and SLAD had 1.5% agarose added.

### Microscopy

A Zeiss Axioplan 2 microscope was used for differential interference contrast (DIC) imaging. The Axiocam MRm camera was used to acquire digital images. For analysis, Axiovision 4.4 software was used. For microscopy imaging, the following strains were used: wild type (PC538) and the *opi1Δ* (PC2847), *tec1Δ* (PC569), *rim101Δ* (PC2953), *ras2Δ* (PC562), and *dig1Δ* (PC3039) mutants. To measure cell length-to-width ratios, images of cells were taken at 100X magnification and the images were imported into the program GIMP (https://www.

gimp.org/downloads/). The measure tool was used to measure the length and width of each cell, and the larger value (length) was divided by the smaller value (width). At least 25 cells were measured per strain and the average ratio was reported with the standard deviation indicated. Budding pattern was visualized and quantified as previously described [101,221]. Fluorescent brightener #28 or Calcofluor white (CFW) was used to stain bud scars. Cells were observed at 100X magnification, and at least 100 buds per strain were counted. Large clumps of cells were excluded to avoid ambiguity. Buds were considered as emerging from either distal, proximal, or equatorial sites and tallied. The number of distal buds was divided by the number of total buds, and this value was reported as a percentage.

## Plate-washing assay

The PWA was used to visualize differences in invasive growth across strains [65,222]. The following strains were used when indicated: wild type (PC538) and the *opi1Δ* (PC2847), *tec1Δ* (PC569), *rim101Δ* (PC2953), *pho85Δ* (PC654), *rtg3Δ* (PC3698), *spt8Δ* (PC4008), *elp2Δ* (PC2976), *sin3Δ* (PC3030), *ras2Δ* (PC562), *pho4Δ* (PC5115), *snf1Δ* (PC560), *slt2Δ* (PC3188), *sch9Δ* (PC5864), *pbs2Δ* (PC2110), *msn2Δ* (PC6094), *tor1Δ* (PC3654), *dig1Δ* (PC3039), *flo11Δ* (PC1029), *bud8Δ* (PC563), and *pea2Δ* (PC551), *tec1Δ rim101Δ* (PC7680), *tec1Δ opi1Δ* (PC7679), *opi1Δ rim101Δ* (PC7681), *ras2Δ tec1Δ* (PC7678), *ras2Δ opi1Δ* (PC7689), and *ras2Δ rim101Δ* (PC7690) mutants. The *rtg3Δ* (PC7677), *sin3Δ* (PC5864), and *pbs2Δ* (PC368) mutants were used on SLAD and +EtOH media and compared to wild type (PC313). Cells were spotted on indicated medium and placed at 30˚ to grow for 7 d unless otherwise indicated. To ensure uniform growth, spots were placed an equal distance to each other and from the edge of the plate. To assess invasive growth, plates were placed under a stream of water, and cells were rubbed gently by hand to remove non-invasive cells. Images of colonies and the invasive scars were taken by ChemiDoc XRS+ molecular imager (Bio-Rad laboratories) under blot/chemicoloric setting with no filter. For close up images of colonys when analyzing complex-colony morphology, a Nikon D3000 digital camera was used.

Invasive growth quantitation has been described [123]. Images were imported into ImageJ (National Institutes of Health, Bethesda, MD, USA; https://imagej.nih.gov/ij/) to remove the background signal. Images were imported into the program GIMP, where images were inverted and adjusted for brightness and contrast. Each image was treated with the same parameters for adjustments. Then, as an adjustment to our previous method, images were imported into the Image Lab 6.0.1 program (Bio-Rad laboratories, https://www.bio-rad.com/en-us/product/image-lab-software?ID=KRE6P5E8Z) for quantification using the round volume tool. Colony borders were outlined to determine area. The circle demarking the colony area was then placed over the image of the washed invasive scar, and the intensity of the invasive scar was determined by the volume tool. Next, the intensity values of the invasive scar was divided by colony area to control for growth differences between strains [intensity / area = invasion]. Invasion values were averaged across three biological replicates. To determine relative invasion, invasion was normalized to wild type, set to 1. Error bars represent standard deviation between trials. Significance was determined using the program GraphPad Prism (commercially available: https://www.graphpad.com/) by one-way ANOVA analysis unless otherwise indicated. Dunnett's multiple comparisons test was used for comparisons of mutants to wild type. Tukey's multiple comparisons test was used to compare mutants to mutants or a mutant across environments. Confidence interval was set to 95%. Reported p-values were adjusted for multiple comparisons. Statistical analyses can be found in *S1–S3 File*.

Invasive growth patterns (plot profiles) were quantified as described [123]. Subtracting out the background signal and adjusting brightness and contrast were performed as stated above.

Images were imported into ImageJ and cropped based on invasive scar size (for example, SGAL: 200 x 200 pixels, SOE: 250 x 250 pixels, YPGAL: 300 x 300 pixels). A box was drawn across the midsection of the invasive scar, from edge-to-edge of the cropped image, with a pixel height of 40. Using the plot profile tool, which measures the grey value for pixels, a plot profile was generated for this region of the invasive scar. Plot profiles were generated along three different axes, which allowed an average value to be reported.

Invasive scar cross sections were performed by spotting equal concentrations of wild-type cells and the *tec1Δ* mutant onto SGAL medium. Plates were incubated for 7 d at 30˚C. Cells were washed off of plates in a stream of water. A razor blade and sterile forceps were used to excise the cells that had become embedded in the agar due to invasive growth. Chunks of agar containing invaded cells were placed on microscope slides to which a coverslip was added. For invasive squashes, gentle pressure was applied to squash the agar, revealing the trapped invaded cells. Cells were visualized by microscopy using the 5X, 10X, and the 100X objectives.

### Measurement of MAPK pathway activity

To analyze **MAPK** pathway activity, the ß-galactosidase (lacZ) assay was performed. The following strains were used: wild type (PC313) and the *tec1Δ* (PC7675), *dig1Δ* (PC7676), *opi1Δ* (PC7674), *rim101Δ* (PC7673), and *ras2Δ* (PC6222) mutants. Cells were spotted on synthetic medium lacking uracil (SD-URA) to maintain selection for plasmids and were incubated for 21 h at 30˚. Cells were then scraped into dH$_2$O and adjusted to the same optical density. Equal amounts of cells were then inoculated in indicated liquid media for 7 h, with shaking, at 30˚. Cells were harvested by centrifugation and stored at -80˚ for at least 30 minutes. The ß-galactosidase (lacZ) assay was then performed as described [223,224] using a transcriptional reporter [p*FRE-lacZ* [217]] as the readout of **MAPK** pathway activity. The assay was performed with three biological replicates. In experiments that compare wild type across environments, values were normalized to YPD medium. In experiments that compare mutants to wild type, values were normalized to wild type. Error bars represent standard deviation between trials. Significance was determined by Student's t-test.

### Quantitative reverse transcription PCR

Quantitative reverse transcription PCR (RT-qPCR) was used to measure the relative expression of the genes *SFG1*, *FLO11*, *NFG1*, *RGD2*, *RPI1*, and *TIP1*. The following strains were used when indicated: wild type (PC538) and the *opi1Δ* (PC2847), *ste12Δ* (PC539), *tec1Δ* (PC569), *rim101Δ* (PC2953), and *ras2Δ* (PC562) mutants. For experiments measuring the expression of *SFG1* and *FLO11*, cells were spotted and grown on YPD or YPGAL semi-solid agar media for 2 d at 30˚ before harvesting. For experiments measuring the expression of *NFG1*, *RGD2*, *RPI1*, and *TIP1*, cells were spotted and grown on YPD (High Glu) semi-solid agar medium for 2 d at 30˚ or inoculated in 5 mL YPGAL liquid medium for 5 h, with shaking, at 30˚ before harvesting. Cells were stored at -80˚ before RNA extractions.

RNA extraction (hot-acid phenol-chloroform extractions), RNA purification [purified using a QIAGEN RNeasy Mini Kit (catalog # 74104), concentration and purity measured with NanoDrop 2000C (Thermo Fisher Scientific)], RNA stability determination (by agarose gel electrophoresis), cDNA generation [generated with iScript Reverse Transcriptase Supermix, Bio-Rad, catalog # 1708841], and RT-qPCR were performed as described in [123]. The Bio-Rad CFX384 Real Time System was used to perform RT-qPCR with iTaq Universal SYBR Green Supermix (Bio-Rad, catalog # 1725121). Primer sequences are shown in *S4 Table*. The $2^{-\Delta \text{Ct}}$ formula was used to calculate relative gene expression [138,225]. Ct was defined as the cycle where fluorescence was statistically significant above background. ΔCt is the difference

in Ct between a target gene and the housekeeping gene *ACT1*. Three biological replicates were used, and average values were reported. Error bars represent standard deviation between trials. Significance was determined by Student's t-test.

The repository YEASTRACT [http://www.yeastract.com/index.php] was used to assess pathway-dependent changes in gene expression.

## Cell adhesion measurements in liquid and semi-solid agar media

For adhesion assays, the following strains were used when indicated: wild type (PC538) and the *opi1Δ* (PC2847), *tec1Δ* (PC569), *rim101Δ* (PC2953), *ras2Δ* (PC562), *dig1Δ* (PC3039), and *flo11Δ* (PC1029) mutants. The adhesion of cells in liquid media was analyzed as described [123]. Cells were grown at 30° for 24 h in 2 ml of specified medium with shaking. Images of groups (clusters) of cells were captured by microscopy at 5X magnification. Images were imported into ImageJ to subtract the background signal and apply a threshold to convert images to binary pixel images, where cells appeared black and the background appeared white. The same parameters were used for each image. The analyze particles tool was then used to measure the average area of cell clusters. The average of three biological replicates was determined and normalized to wild-type cells, which was set to 1. Error bars represent the standard deviation between trials. Significance was determined by Student's t-test.

Cell adhesion on agar media was quantified as described [123]. Cells were spotted onto semi-solid agar medium (as indicated) and incubated at 30° for 7 d. Cells were harvested and resuspended in dH$_2$O in 50 ml conical tubes. Tubes were inverted vigorously 10 times, and the contents of the tube were poured into a petri dish, where groups of adherent cells formed "adhesive particles" visible by eye. Particles were imaged by ChemiDoc XRS+ molecular imager under blot/chemicoloric setting with no filter. Images were digitally cropped by ImageJ using the circle tool around the edge of the Petri dish and background signal was subtracted. The same threshold was applied to each image to generate binary pixel images, where particles appeared black and the background appeared white. Using the analyze particles tool, total particle area was measured to represent total adhesion. Total adhesion was normalized to wild type, which was set to 1, and displayed as relative total adhesion. The average of three biological replicates was determined, where error bars represent the standard deviation between experiments. Significance was determined by Student's t-test.

## Plastic adhesion assay

Measurement of the adhesion of cells to plastic was performed as described [134]. The following strains were used: wild type (PC538) and the *opi1Δ* (PC2847), *tec1Δ* (PC569), *rim101Δ* (PC2953), *ras2Δ* (PC562), *dig1Δ* (PC3039), and *flo11Δ* (PC1029) mutants. Cells were spotted onto semi-solid agar medium (as indicated) and incubated at 30° for 7 d. Cells were removed from the community surface with a toothpick, resuspended in dH$_2$O, and adjusted to the same optical density. Equal amounts of cells were added to polystyrene wells (Falcon Microtest Tissue culture plate, 96 Well) and incubated for 4 h at 30°. Crystal violet dye (DIFCO) was then added for 20 min, and wells were washed equally and imaged with a Nikon D3000 digital camera. Quantification was performed by ImageJ analysis as described [123]. Each well was digitally cropped by the circle tool, equally from the center of the well. An equal threshold was used to convert images to binary pixel images, where adherent stained cells appear black against a white background. The analyze particle tool was used to measure the total area of adherent cells. Values were normalized to wild type that was set to 1. The experiment was performed in duplicate, and error was displayed as standard deviation from biological replicates.

## Supporting information

**S1 Fig. Full survey of network mutants by the PWA. A)** PWA on indicated media. First column, cells before wash, second column, inverted images of scars after wash, bar, 0.5 cm. Quantification in *S2 Fig*. **B)** PWA on +KCl medium; details in panel A. Only the *dig1Δ* mutant invades in this environment. **C)** All replicates on all media for wild type and the mutant strains were plotted for invasion versus colony size (mm$^2$). No correlation between invasion and colony size occurred ($R^2 = 0.0004$).
(PDF)

**S2 Fig. Quantification of the PWA. A)** PWA; Levels of relative invasion to wild type in indicated media, with wild-type values set to 1. Asterisk, p-value $\leq 0.05$, compared to wild type. (Images in *S1A Fig*) **B)** Levels of relative invasion for the *dig1Δ* mutant, with wild-type values set to 1; Asterisk, p-value $\leq 0.05$, compared to wild type by Student's t-test.
(PDF)

**S3 Fig. Invasive growth network ranking. A)** Ranking of network components per their regulatory role of invasive growth in environments where invasive growth occurs (11 out of 12 environments, excludes +KCl). **A)** The average relative invasion compared to wild type (set to 1) was calculated across all environments for network mutants. Error represents standard deviation. **B)** Total number of environments a network mutant met a threshold of decreased invasion relative to wild type for indicated thresholds: $\leq 75\%$, $\leq 50\%$, $\leq 25\%$, $\leq 20\%$, $\leq 15\%$, or $\leq 10\%$ of wild-type invasion. Each threshold was totaled independently. The pathways were ranked in order using all thresholds (# of thresholds).
(PDF)

**S4 Fig. Growth of wild-type cells and mutants lacking the major regulatory pathways that control invasive growth.** Wild-type cells and the indicated mutants were grown in the indicated media in liquid culture for 16h at 30° by shaking. Cells were washed once in water, and growth was measured by OD at 600nm. For each condition, wild-type values were set to a value of 1. The average of three replicates is reported. Error is reported as standard deviation.
(PDF)

**S5 Fig. Plot profile analysis. A)** PWA. Inverted images of invasive scars shown. Bars, 0.5 cm. **B-D)** Plot profile of invasive growth across invasive scars. Strains and media are as indicated; examples are shown in panel A. X-axis, distance (in pixels); Y-axis, pixel intensity.
(PDF)

**S6 Fig. Complex-colony morphology does not correlate to invasive growth. A)** Complex-colony morphology analysis. Images of complex-colony morphology are shown. Images are also shown in *S1 Fig*. Bar graphs, levels of relative invasion, with wild type values set to 1. Black asterisk, p-value $< 0.05$, compared to wild type. Purple asterisk, p-value $< 0.05$, comparing the mutants to each other by Student's t-test. Invasive scar images can be found in *S1 Fig*. Relative invasion values are also shown in *S2 Fig*. The *rtg3Δ* mutant showed decreased invasion on YPD and YPGAL media compared to wild type yet had a similar complex-colony morphology pattern. The *flo11Δ* mutant showed increased invasion from YPD to YPGAL media with no change in its complex-colony morphology pattern. Moreover, the *flo11Δ* mutant showed higher invasion but lower complex-colony morphology than the *rtg3Δ* mutant on YPGAL. **B)** Model of above and below surface communities depicting the role of **RTG**. CCM, complex-colony morphology.
(PDF)

**S7 Fig. PWA over time on YPGAL and SD media. A)** PWA. Strains were spotted on YPGAL for the indicated number of days (2 d, 3 d, 7 d, and 21 d). Top row, cells before wash, bottom row, inverted images of scars after wash, bar, 0.5 cm. **B)** Same as panel A, except on SD medium. **C)** We found a strong temporal role on SD medium but not YPGAL medium for the *dig1Δ* mutant. Levels of relative invasion between wild type and the *dig1Δ* mutant on indicated medium, with wild type values set to 1. Left, YPGAL, right SD. Asterisk, P-value ≤ 0.05, compared to wild type. **D)** Levels of relative invasive growth versus relative **MAPK** pathway activity in wild-type cells. **E)** Levels of invasive growth on YPGAL or SD media over 21 d for wild type (black) and the *dig1Δ* (purple) and *tec1Δ* (green) mutants.
(PDF)

**S8 Fig. PWA of double mutants. A)** PWA; First column, cells before wash, second column, inverted images of scars after wash, bar, 0.5 cm. **B)** Levels of relative invasion to wild type, with wild type values set to 1. Asterisk, p-value ≤ 0.05, compared to wild type. **C)** p-values of double mutants compared to single mutants. Red highlights, p-value ≤ 0.05.
(PDF)

**S9 Fig. Network mutants play different roles to bring about similar wild-type phenotypes. A)** PWA of wild type on SLPD and SD media, bar, 0.5 cm. Images are repeats from *S1A Fig*. Bar graph, levels of relative invasion to SLPD, with SLPD values set to 1. Quantification values are repeated from **Fig 1B**, except in relative terms. **B)** PWA on SLPD and SD media. Images are repeats from *S1A Fig*. **C)** Levels of relative invasion to wild type, with wild-type values set to 1. Quantification values are from *S2 Fig*. Asterisk, p-value ≤ 0.05, comparing one strain to itself between SLPD and SD media by Student's t-test.
(PDF)

**S10 Fig. Cell adhesion in liquid cultures. A)** Cell adhesion in liquid cultures. Cells were grown in indicated media and imaged by microscopy at 5X magnification, bar, 200 μm. **F)** Quantification of cell clusters. Asterisk, p-value ≤ 0.05, compared to wild type.
(PDF)

**S11 Fig. Pathways show variation in complex-colony morphology and cell-cell adhesion on solid surface environments. A)** Complex-colony morphology, or patterning on the surface of a community of cells. The more ruffly a complex-colony morphology the more adhesion between cells. Images of colony after 7 d of growth on indicated media, bar, 0.5 cm. Wild-type cells show strong ruffling on YPD and YPGAL media and weaker but still increased complex-colony morphology compared to the *flo11Δ* mutant on SOE and SD media. The *flo11Δ* mutant exhibited a smooth pattern in all environments. All mutants tested showed reduced complex-colony morphology compared to wild type. Compared to the *flo11Δ* mutant the following mutants showed increased complex-colony morphology on indicated media: *tec1Δ* mutant on YPGAL medium; *ras2Δ* mutant on YPD medium; *rim101Δ* mutant on SOE medium; *opi1Δ* mutant on YPD medium; *dig1Δ* mutant on all media. **B)** Total cell adhesion within a colony. Images of adherent cells from the colony surface are seen as black particles. Bar graphs, levels of relative total adhesion compared to wild type, with wild-type values set to 1. Asterisk, P-value ≤ 0.05, compared to wild type. The *flo11Δ* mutant exhibited no detectable adhesion within the colony on any environment. Wild-type cells showed cell-cell adhesion on all four environments. The *tec1Δ* and *ras2Δ* mutants showed increased adhesion on YPGAL medium compared to the *flo11Δ* mutant.
(PDF)

**S12 Fig. Plastic adhesion assay.** Plastic adhesion is a medically relevant phenotype because pathogenic yeasts, like *Candida albicans*, will adhere to medical devices and plastics in hospital settings. Images of stained cells adhering to a polystyrene plastic 96-well plate. Bar graph, quantification of relative plastic adhesion to wild type, with wild type values set to 1.
(PDF)

**S13 Fig. MAPK regulates target genes in one environment but not another.** Targets of the MAPK pathway, *NFG1*, *RGD2*, *RPI1*, and *TIP1* identified previously [100], were regulated by **MAPK** in one environment but not another. **A)** RT-qPCR analysis of mRNA levels for indicated genes between wild-type and the *ste12Δ* mutant (a **MAPK** pathway mutant equivalent to *tec1Δ*) in YPGAL medium. Wild-type values were normalized to *ACT1* expression and set to 1. Asterisk, p-value ≤ 0.008, compared to wild type. RT-qPCR data for YPGAL comes from [100]. **B)** Same as panel A, except on YPD (High Glu) medium. **C)** These genes also show different changes in expression between the two environments. RT-qPCR analysis of mRNA levels between YPGAL and YPD (High Glu). Wild-type values were normalized to *ACT1* expression. YPD (High Glu) values were set to 1. Asterisk, p-value ≤ 0.005, comparing YPGAL to YPD (High Glu).
(PDF)

**S14 Fig. Filament-like structures form in the RAS, RIM101, and OPI1 mutants, but not the MAPK pathway mutant.** Microscopy images at 20X magnification of invasive scars on indicated media, bar, 50 μm. Each strain, except the *tec1Δ* mutant, showed the capability of producing a filament-like structure in each environment tested.
(PDF)

**S15 Fig. Cross section analysis.** The PWA was performed on SGAL medium. A small, thin section of the invasive scar was cut and placed on its side to view the cross section. **A)** Colored image of invasive scar cross section by a Nikon D3000 digital camera, bar, 0.5 cm. **B)** Microscopy images of invasive scar cross section. 5X magnification, bar, 500 μm. Red arrows, invading cells. 10X magnification, bar, 100 μm. Red arrows, same invading cells in 5X image. 20X magnification, bar, 100 μm. Red arrows, same invading cells in 5X and 10X images. For the 10X and 20X magnification images, 3 focal planes were imaged of the same cells (Focus 1/2/3).
(PDF)

**S16 Fig. Invasive squashes to identify morphologies of cells undergoing invasive growth.**
**A)** Examples of invaded cells for wild-type cells and the *tec1*D mutant. Cells were visualized by microscopy at the 100X objective. Bar, 20 microns. **B)** Raw data of invasive squashes of wild-type cells and the *tec1*D mutant. Cells were visualized by microscopy at the 5X, 10X, and 100X objectives. Bars, 5, 10, and 20 microns as indicated. Several examples are shown from different squashes.
(PDF)

**S1 File. Statistical analysis of pairwise comparisions for each strain across environments.**
(XLSX)

**S2 File. Statistical analysis of pairwise comparisions for each strain compared to wild type.**
(XLSX)

**S3 File. Statistical analysis of pairwise comparisions for each mutant compared to each mutant.**
(XLSX)

**S1 Table. Media used in this study representing diverse environments.**
(PDF)

**S2 Table. References for the cross regulation of pathway components.**
(PDF)

**S3 Table. Yeast strains used in this study.**
(PDF)

**S4 Table. RT-qPCR primers used in this study.**
(PDF)

## Acknowledgments

Thanks to Hiten Madhani (University of California, San Francisco, CA) for the plasmids. Thanks to Laura Rusche, Trevor Krabbenhoft, and Denise Ferkey (University at Buffalo, Buffalo, NY) and Christian Landry (Universite Laval, Quebec City, Quebec) for helpful comments and insight. Thanks to Nathan Backenstose, Alexander Bowitch, Aditi Chaubey, Bea Gonzalez, and past and current laboratory members for helpful comments and support. Thanks to Kate Vandermeulen for her support.

## Author Contributions

**Conceptualization:** Matthew D. Vandermeulen, Paul J. Cullen.

**Formal analysis:** Matthew D. Vandermeulen.

**Funding acquisition:** Paul J. Cullen.

**Methodology:** Matthew D. Vandermeulen.

**Project administration:** Paul J. Cullen.

**Resources:** Paul J. Cullen.

**Supervision:** Paul J. Cullen.

**Validation:** Matthew D. Vandermeulen.

**Visualization:** Matthew D. Vandermeulen.

**Writing – original draft:** Matthew D. Vandermeulen.

**Writing – review & editing:** Matthew D. Vandermeulen, Paul J. Cullen.

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
