## [Decision Letter · Decision Letter 0]

20 Oct 2021

Dear Dr Cullen,

Thank you very much for submitting your Research Article entitled 'Gene by Environment Interactions Reveal New Regulatory Aspects of Signaling Network Plasticity' to PLOS Genetics.

The manuscript was fully evaluated at the editorial level and by three independent peer reviewers. The reviewers appreciated the attention to an important problem, but two raised substantial concerns about the current manuscript. Based on the reviews, we will not be able to accept this version of the manuscript, but we would be willing to review a revised version. We cannot, of course, promise publication at that time.

Reviewer 2 has two main concerns and provided suggestions as to how to improve the manuscript. These comments include quantification of the observations made and statistical analyses that would support or refute additional potential hypotheses that could be proposed. The second main comment regards delving deeper into the interesting observations and providing data from molecular work. If you can provide such examples, I agree that they would strengthen this manuscript. Please explicitly address these main concerns by experiment and analyses, or (especially for the molecular work) explain why you deem this beyond the scope of this work. Suggestions of Reviewer 3 regarding citations and claims of novelty should be addressed by re-writing. 

Should you decide to revise the manuscript for further consideration here, your revisions should address the specific points made by each reviewer. We will also require a detailed point-for-point list of your responses to the review comments and a description of the changes you have made in the manuscript.

If you decide to revise the manuscript for further consideration at PLOS Genetics, please aim to resubmit within the next 60 days, unless it will take extra time to address the concerns of the reviewers, in which case we would appreciate an expected resubmission date by email to plosgenetics@plos.org.

[LINK]

We are sorry that we cannot be more positive about your manuscript at this stage. Please do not hesitate to contact us if you have any concerns or questions.

Yours sincerely,

Michael Freitag

Associate Editor

PLOS Genetics

Gregory P. Copenhaver

Editor-in-Chief

PLOS Genetics

Reviewer's Responses to Questions

**Comments to the Authors:**

Reviewer #1: This manuscript by Vandermeulen and Cullen takes advantage of the invasive growth phenotype in budding yeast to quantify gene-environment interactions and how phenotypic plasticity can be mediated through condition-specific signaling. The authors examined mutants lacking main regulators of invasive growth across multiple environments, which allowed them to determine their relative importance under different conditions. In addition to the pathways known to play major roles in regulating invasive growth, the authors found that the transcriptional regulator Opi1 plays an even larger role under most conditions. The authors tested various double mutants under different conditions, which revealed that pathways can act additively, redundantly, or even opposingly depending on the environment. Finally, the authors measured in mutants the gene expression of target genes ultimately responsible for different aspects of invasive growth, ultimately allowing them to begin to connect different pathways to expression output.

Altogether, I think this manuscript rather thoroughly describes the diverse ways that different signaling pathways can interact leading to both regulatory and organismal phenotypic plasticity. I do have some comments and suggestions, largely to clarify for the reader.

1) I think it would be helpful to have the first figure show at least the abbreviated major pathways (Rim101, Pka/Ras, MAPK) and how they connect to major output genes and cellular phenotypes relevant to invasive growth (FLO11 - adhesion, BUD8 - distal budding, PEA2 - elongation).

2) Please consider changing the color scheme for red/green heat maps for the sake of any colorblind colleagues (blue/red should work).

3) While I appreciated how the authors used the ranking of mutational effects to determine which pathways played the largest roles under different environments, the authors may also consider hierarchical clustering by both mutant and environment (or just calculating correlation coefficients). In particular, clustering by environment may help determine which signals are being used by which pathways (e.g. nitrogen level vs. carbon source), and it may help to determine the environmental trigger that causes certain mutants to no longer have changes in reaction norms.

4) Minor point: how was the order of mutants determined in Figure 1--it’s not exactly the same as S3.

5) This may be coincidental, but the opi1 mutant seems to be the phenotypic mirror image of dig1. Reynolds 2020 speculates that Opi1 is a repressor that is indirectly causing FLO11 activation. Could it be that Opi1 is a repressor for Dig1? There may be some transcriptomic datasets that could help answer that question.

6) For the double mutant analysis, lack of significance in a t-test is not really the same as testing for equivalency, and it may be inherently difficult with the invasive growth assay to determine non-additivity for mutants with already major defects. I think the interpretation of interacting pathways like Ras should be toned down unless there is other evidence for cross talk. Otherwise, I think testing expression of reporter genes like FLO11 in single and double mutants may be more sensitive for identifying conditions where pathways are acting in parallel or are interacting.

7) I’m misunderstanding something in Figure 4 comparing “A” and “C.” The order of pathway activity in “C” does not match the order in “A.” Likewise, not sure where the order in “D” for invasive growth is coming from.

8) Figure 5H: Consider changing to have the arrows all point in the same direction like Figure 3C. For gene expression, I instinctively associated up arrows with up-regulation and down arrows with down-regulation.

Reviewer #2: This is an interesting paper that uses gene-by-environment analysis of invasive growth in yeast to identify conditional gene-gene and gene-phenotype relationships within the signaling network that controls the phenotype. This work has the potential to be of interest to a wide group of researchers interested in signaling network dynamics and the plasticity of networks under different environmental conditions.

I have two major concerns with the paper, one suggestion for increasing the impact of the work and a number of minor issues.

First major concern: I am not convinced that the invasive growth measurements fully control for confounding effects of growth. Invasive growth is measured as scar-intensity/colony area. Unless this ratio stays the same throughout colony growth then you could imagine the following scenario: Wild-Type has an invasion value of 2 at day-2 (small colony), value of 3 at day-5 (medium colony) and value of 4 at day 7 (large colony). Deleting gene X has an effect only on growth, so that colonies follow the same trajectory as WT but with a delay, so that at day 5 they have an invasion value of 2 (small colony) and at day 7 an invasion value of 3 (medium colony). Using the authors’ approach, this would suggest that deletion of gene X causes lower than WT invasion at days 5 and 7 (ratios of 2/3 and ¾), when all that was really happening was reduced growth, with no effect on invasion behavior (relative to colony size). This line of thought suggests that normalization between genotypes or conditions should be done between colonies of the same size. Can the authors comment on this concern?

Second major concern: I have a major concern with the statistical analyses. The research presented is exploratory and involves a large number of statistical tests (all t-tests, I believe), both explicitly presented in the figures and implicitly in the fact that the examples that are illustrated are derived from a much larger number of possible comparisons, with the choice of what to present likely driven by what appeared most significant. I can find no mention in the paper of correcting p-values for multiple hypotheses. If I am correct that no correction for multiple hypotheses was undertaken, then I am skeptical that many of the fairly modest effects discussed are actually statistically significant.

Increasing the impact: The paper does a thorough job of illustrating that gene-gene and gene-phenotype relationships can change between environments. There is also some illustration of how changes in gene expression of a small number of effector genes accompany these changes in environment and genotype. To make this paper of more general interest, the authors need more molecular data explaining HOW an environmental change changes gene-gene and gene-phenotype relationships, perhaps exploring expression levels and phosphorylation states of more genes in the signaling network, rather than just downstream effectors.

Minor Points:

The authors need to be more precise about the meaning of terms used, particularly as relating to “regulating” the signaling network. It is often unclear whether this means changing the network graph, or producing different outputs from a stable network because of different environmental inputs.

P5. “Phenotypic changes based on the environment are referred to as Gene (or Genotype) by Environment Interactions (GEIs)”. This definition of GEI is incorrect (it does not include the effect of different genes/genotypes).

P9. “The fact that invasive growth regulatory pathways show different roles depending on environment is surprising, because the prevailing view is that the major pathways play an equal role, in that their loss causes a complete reduction in invasive growth” – I think this is a straw man and that this view is not generally held. In the absence of a reference clearly arguing that the role of the different pathways is equal, the authors should remove this claim.

P9. The ranking exercise the authors undertake is interesting, but they need to be clearer about its limitations. The set of conditions that were tested is only a very small subset of all possible environmental conditions, therefore the ranking is specific only for the set of conditions used and cannot be generalized to an absolute ranking of the pathways. In addition, describing Opi1p as the “main regulator of invasive growth” implies ‘necessary and sufficient’ (at least to me) perhaps something more cautious, like “strongest single regulator” would be better.

P9. “Invasive growth was also examined to identify the roles pathways and protein complexes play in regulating changes to invasive growth that occur when transitioning from one environment to another”. The authors should rewrite this as no transitions actually occurred, instead invasion was simply compared between conditions for different mutants. Similarly, on P10 “when transitioning between certain environments”.

P10. I am not sure about the utility of Figure 1E. I have concerns about normalizing to YPD for the reaction norms (Figure 1E) or normalizing to any other condition (Figure S4). In terms of this study there is nothing special about YPD (or any other condition) that makes it an appropriate “base condition”, instead it is simply one of several environments tested. For deletion strains where the normalization condition is an outlier, this will distort their entire profile across the remaining environments. This issue affects the whole discussion on P10. Instead, it would be safer to compare rows directly from Figure 1C to make arguments about how the importance and direction of effect of different genes (relative to WT) changes in different conditions. This is because WT is an appropriate baseline to compare the other genotypes to.

P12 The second paragraph, beginning “The invasive growth network” makes contradictory statements about the set of genes regulating invasive growth vs complex colony morphology. I think the authors are arguing that the same gene network regulates both phenotypes, but that the relative importance and role of genes/pathways in the network differs between the above-surface and below-surface environments (presumably through different inputs into the signaling network). The authors need to clarify what they mean here.

P12. I’m not sure it is possible to determine roles in “initiating” and “maintaining” invasive growth using the results in Figure 2F. These results look consistent with a simple reduced rate of invasive growth in all of the mutants.

P13. Why does the fact that the sum of the mutant invasion scores is greater than 1 suggest redundancy?

P13 Line304. I see very few instances where a pathway is required for invasion (OPI1 on WL medium, is one example) and many situations where pathways are partially dispensible.

P15. “Intriguingly, the environment where RTG played a negative role, PHO85p played a positive role and vice versa”. Deletion of either reduces invasion in YPD, so this is only true for SD vs SLPD.

P16. It would be more informative to see a scatterplot of MAPK activity against (WT) invasive growth in the different environments tested rather than trying to compare Figures 4C and 4D.

P16 The use of the Tec1 and Dig1 deletions to define upper and lower invasive growth boundaries in Fig 4D is not explained.

P20 +21. I am not sure that expression levels of SFG1 are a good proxy for FLO11-independent adhesion mechanisms, particularly because SFG1 also regulates FLO11.

P23. Describing OPI1 as a “top regulator” implies it is upstream of all the pathways, not that it has the single strongest effect on the phenotype.

P26 What does “mutant phenotype analysis” refer to that is different from GEI?

P26 The sentence beginning “GEI analysis…” is very opaque and the authors should clarify their meaning

Reviewer #3: -Overall very interesting dataset that will be of interest to this community. I believe the data merit strong consideration for publication, if care is taken to correct issues in the presentation of that data. In particular, I felt that the paper was careless in its claims of novelty for some of its findings. My concerns are further spelled out below.

-Strong statements throughout about prior knowledge in the field. I don't agree that all the reported findings are new, for instance some of what they claim is new is spelled out in the review Bruckner 2012, which is cited. A few example lines are below. More broadly, statements like "How networks are regulated to produce different phenotypes in different environments is not well understood" in the abstract could use justification. Much work in molecular biology has focused on how pathways respond to environmental cues.

"These results indicate that invasive growth is not binary (ON/OFF) but occurs in a phenotypic spectrum based on the environment." [182]

"the prevailing view is that the major pathways play an equal role, in that their loss causes a complete reduction in invasive growth" [201]

"In any case, this remarkable finding demonstrates that the same phenotype can arise through the action of pathways that have opposing roles in different settings." [Line 342]

-Definition of GEI in line 111 seems off. Shouldn't that be about variant genotypes interacting with the environment, not just phenotypes changing due to environment?

-Further, I am not sure I came away clear on what GEI analysis is. There is not a clear definition of this as an approach. Line 156 ("Coupling GEI analysis to conventional mutant analysis") implies that the authors are treating GEI analysis as something other than exposing mutants to different environments, which is how I would define GEI analysis.

-The use of "reaction norms" starting at line 220 could use expansion. Materials and methods were insufficient in describing what this is or what it can tell you that is distinct from a standard normalization procedure.

-In line 253, "invasion scar" is used for the first time without a definition. Could go in the introduction when defining the phenotype.

-Work from the Ehrenreich lab (and others) has shown that mutations related to invasion phenotypes are strongly influenced by genetic background. It would be good to point this out, as some of the findings, like the dominance of the OPI pathway, may be background-specific. A few lines on this are in the figure 1 legend, but should be located somewhere in the main text instead and would benefit from more discussion.

-The mix of transcription factors and upstream regulators may also impact the results regarding the ranking of pathways. When Ras2 is knocked out, its downstream regulators are still present, whereas with OPI, the actual TF is gone. Adding in some additional mutants in genes in different locations withing these signaling pathways would be great, if possible. At the least discussion of the possible impacts of these choices is essential in my view.

-Many interpretive statements would benefit from being moved to the discussion, such as line 372 "Perhaps other pathways operate at low overall levels as a safeguard to curb pathway activity. It is clear at least for the ERK pathway that overactivation can lead to cancer and other diseases."

-Figure 6b on round cell invasion: I'm not sure I see it like the authors. The cross section looks like a mix of elongated and round cells. Could it be that only some cells elongate to invade, and some incidental round cells hang around after washing due to expression of flocculins?

-Two papers that may be of interest came to mind while reading this. I believe they may have useful information for further interpreting your results:

1.

1. Reynolds, T. B., Jansen, A., Peng, X. & Fink, G. R. Mat Formation in Saccharomyces cerevisiae Requires Nutrient and pH Gradients. Eukaryot Cell 7, 122–130 (2008).1.

2. Váchová, L. & Palková, Z. How structured yeast multicellular communities live, age and die? FEMS Yeast Res 18, (2018).

-Again, this is a very exciting dataset and I hope that these comments are constructive.

**Have all data underlying the figures and results presented in the manuscript been provided?**

Reviewer #1: Yes

Reviewer #2: Yes

Reviewer #3: Yes

PLOS authors have the option to publish the peer review history of their article (what does this mean?). If published, this will include your full peer review and any attached files.

Reviewer #1: No

Reviewer #2: No

Reviewer #3: No

---

## [Editor Report · Decision Letter 1]

9 Dec 2021

Dear Dr Cullen,

We are pleased to inform you that your manuscript entitled "Gene by Environment Interactions Reveal New Regulatory Aspects of Signaling Network Plasticity" has been editorially accepted for publication in PLOS Genetics. Congratulations!

Your revised version and the point-for-point response letter addressed all comments or concerns of the previous reviewers. After carefully reading your extensive documentation, I found it unnecessary to send the manuscript out for a second round of reviews, as you addressed the concerns of all reviewers more than adequately; you almost completely followed their advice to modify text, or include additional figures and data. You carried out the requested additional experiments (all reviewers), and provided the requested additional statistical analyses to address the possibility of addressing additional, alternative hypotheses (Reviewers 1 and 2). Concerns by Reviewer 2 about varying growth effects are now addressed by Fig S4. You adapted blue/red color schemes to avoid green/red in figures. You updated your citations and modified wording as needed, as was requested by Reviewer 3. You also went beyond the reviewers’ comments to improve the manuscript, as indicated in the last section of your response letter. As the reviewers did, we applaud your thorough approach to identifying gene-phenotype relationships in a complex signaling environment, and believe that your work will be interesting to a wide audience working on network dynamics and phenotypic plasticity under changing environmental conditions.

Yours sincerely,

Michael Freitag

Associate Editor

PLOS Genetics

Gregory P. Copenhaver

Editor-in-Chief

PLOS Genetics

Comments from the reviewers (if applicable):

**Data Deposition**

http://datadryad.org/submit?journalID=pgenetics&manu=PGENETICS-D-21-01193R1

**Press Queries**

---

## [Editor Report · Acceptance letter]

30 Dec 2021

PGENETICS-D-21-01193R1 

Gene by Environment Interactions Reveal New Regulatory Aspects of Signaling Network Plasticity 

Dear Dr Cullen, 

We are pleased to inform you that your manuscript entitled "Gene by Environment Interactions Reveal New Regulatory Aspects of Signaling Network Plasticity" has been formally accepted for publication in PLOS Genetics! Your manuscript is now with our production department and you will be notified of the publication date in due course.

With kind regards,

Zsofia Freund

PLOS Genetics

On behalf of:
